# A Hierarchical Nearest Neighbour Approach to Contextual Bandits

**Stephen Pasteris**                                                                *spasteris@turing.ac.uk*
*The Alan Turing Institute, London, UK*

**Madeleine Dwyer**                                                                *mdwyer@turing.ac.uk*
*The Alan Turing Institute, London, UK*

**Chris Hicks**                                                                       *c.hicks@turing.ac.uk*
*The Alan Turing Institute, London, UK*

**Vasilios Mavroudis**                                                          *vmavroudis@turing.ac.uk*
*The Alan Turing Institute, London, UK*

**Reviewed on OpenReview:** *https://openreview.net/forum?id=4bJMIrI5oX*

## Abstract

In this paper we consider the contextual bandit problem in metric spaces of bounded diameter. We design and analyse an algorithm that can handle the fully adversarial problem in which no assumptions are made about the space itself, or the generation of contexts and losses. In addition to analysing our performance on general metric spaces, we further analyse the important special case in which the space is euclidean, and furthermore analyse the i.i.d. stochastic setting. Unlike previous work our algorithm is adaptive to the local density of contexts and the smoothness of the decision boundary of the comparator policy, as well as other quantities. Our algorithm is highly efficient - having a per-trial time polylogarithmic in both the number of trials and the number of actions when the dimensionality of the metric space is bounded. We also give the results of real world experiments, empirically demonstrating the performance of our algorithm.

## 1 Introduction

We consider the contextual bandit problem in metric spaces. In this problem we have some (potentially unknown) metric space of bounded diameter. We assume that we have access to an oracle for computing distances between pairs of points. On each trial $t$ we are given a *context* $x_t$ belonging to the space, and must choose an *action* $a_t$ before observing the loss/reward generated by that action. In this paper the contexts are considered implicit and we define $\Delta_{s,t}$ to be the distance between $x_s$ and $x_t$.

This problem has been well-studied in the i.i.d. stochastic case where contexts and losses are drawn i.i.d. from a fixed distribution. Whilst some works (e.g. Slivkins (2009), Reeve et al. (2018), Perchet & Rigollet (2011) and references therein) solve the problem directly, others (e.g. Agarwal et al. (2014) and Foster & Rakhlin (2020)) reduce the stochastic bandit problem to supervised learning and hence utilise a classification or regression oracle. Although Slivkins (2009) also study a non-stochastic version, they place strong constraints on the generation of the contexts and losses and do not bound the regret with respect to an arbitrary policy. In this paper we consider the fully adversarial problem in which no assumptions are made at all about the metric space, context sequence, or loss sequence. As far as we are aware the first result for the fully adversarial problem was given by Pasteris et al. (2023) who give an algorithm that we refer to as NN and bound the cumulative loss with respect to that of any policy (i.e any mapping of the trials/contexts to actions). This regret bound is good when the contexts partition into well separated clusters and the policy

is constant on each cluster. However, the bound is poor when there exist many contexts lying close to the decision boundary of the comparator policy. In order to (partially) rectify this, they proposed using *binning* as a preprocessing step. The process of binning involves, at the start of each trial $t$, finding an approximate nearest-neighbour $x'$ of the current context $x_t$ in the set of all contexts seen so far. If the distance from the $x_t$ to $x'$ is no greater than some constant called the *binning radius* then $x_t$ is replaced by $x'$. The paper gave no method to automatically tune the binning radius, whose optimal value depends on the context density and the smoothness of the decision boundary of the comparator policy. Even if one were to choose the optimal binning radius, it would be constant across the whole space, meaning that the algorithm does not locally adapt to the density and smoothness. This inability to adapt can result in an extreme loss of performance: as we show in Section 4.4. In this paper we fully rectify this problem, designing a new (but related) algorithm HNN which (implicitly) automatically adapts to the local density, smoothness and other quantities. In Section 4.4 we give an example of how we dramatically improve over NN (when NN has the optimal binning radius).

As in NN, HNN utilises the algorithm CBNN of Pasteris et al. (2023) which receives, on each trial $t > 1$, some previous trial $p_t \in [t-1]$ (which is a *similar* trial and is denoted $n(t)$ in Pasteris et al. (2023)). For NN, $p_t$ was chosen such that $x_{p_t}$ is an approximate nearest neighbour of $x_t$ in the set $\{x_s \mid s \in [t-1]\}$. In this paper we use a different choice of $p_t$ which we call an approximate *hierarchical nearest neighbour*. For approximate hierarchical nearest neighbour we will construct, online, a partition of the trials seen so far into different *levels*. On each trial $t$ the algorithm then finds an approximate nearest neighbour on each level. $p_t$ will then be chosen, in a specific way, from these approximate nearest neighbours. As HNN is based on CBNN, it inherits its extreme computational efficiency: having a per trial time complexity polylogarithmic in both the number of trials and number of actions when the dataset has an aspect ratio polynomial in the number of trials (which is enforced by binning) and our metric space has bounded doubling dimension. Due to the extreme complexity of CBNN, its description is outside the scope of this paper. Instead, inspired by the use of standard belief propagation in Pasteris et al. (2023), we detail how the *online belief propagation* of Delcher et al. (1995) can be used instead, giving a slight decrease in efficiency.

We note that the process of inserting a given trial/context into our data-structure is very similar to the process of constructing a new bin in Slivkins (2009). However, the algorithms themselves are very different and they are analysed in very different ways (our analysis being far more involved than that of Slivkins (2009)). Nevertheless, we cite Slivkins (2009) as an inspiration for this paper.

## 1.1 Additional Related Work

The bandit problem Lattimore & Szepesvári (2020) was first studied by Robbins (1952) in the non-contextual case, where we receive no side information at the beginning of each trial - we simply select an action and see it's loss. The first algorithms for this simple problem only worked in the stochastic case Lai & Robbins (1985); Agrawal (1995); Auer et al. (2002a) in which, for each action, the loss associated with that action on any trial is drawn from a fixed probability distribution. The constraint that this probability distribution is fixed is a strong one, and for many applications is not satisfied. The first algorithm to work when this constraint is not satisfied was the EXP3 algorithm of Auer et al. (2002b), which is a modification of the classic HEDGE algorithm of Freund & Schapire (1997) for the full-information variant of the problem (where the losses associated with all actions are revealed after every trial). The EXP3 setting is fully adversarial, in which (apart from the losses being bounded) no assumptions are made on the generation of the loss vectors.

The paper Auer et al. (2002b) also introduced the EXP4 algorithm, which generalises the EXP3 algorithm to allow for expert advice (upon which action to take) to be given at the start of each trial. EXP4 gives bounds on the cumulative loss relative to that of following the advice of any particular expert. This experts framework can be viewed as contextual bandits with an implicit context. i.e. the experts are functions mapping the context space to the action set - on each trial a context is observed and each expert gives to us the action that it associates with that context. However, given that we know all contexts a-priori, being able to bound the cumulative loss with respect to that of any mapping of contexts to actions requires an exponential number of experts: meaning that EXP4 takes a time exponential in the number of trials $T$.

As noted in Herbster et al. (2021), when given a tree structure over the contexts and a specifically crafted initial weighting (defined by the tree) of the possible experts, we can use *belief propagation* Pearl (1982) to implement Exp4 in a per trial time of $\mathcal{O}(KT)$ where $K$ is the number of actions. When a parameter known as the *learning rate* is tuned appropriately, the cumulative loss of this algorithm is bounded above by that of any mapping of contexts to actions plus $\tilde{\mathcal{O}}(\sqrt{\Phi KT})$, where $\Phi$ is the cut-size of that mapping with respect to the tree-structure. Given any graph over the contexts, by first sampling a random spanning-tree Herbster et al. (2008); Cesa-Bianchi et al. (2010), we can utilise this tree-structured algorithm to obtain loss bounds with respect to labelings of the initial graph by actions. However, the per-trial time complexity of $\mathcal{O}(KT)$ can be prohibitive, so Herbster et al. (2021) utilised the methodology of specialists Freund et al. (1997); Koolen et al. (2012) in order to develop efficient algorithms (called the GABA algorithms) for this problem: one based on belief propagation (with a time complexity of $\mathcal{O}(K \ln(T))$) and one based on the specialist set described in Herbster & Robinson (2018) (with a per-trial time complexity of $\mathcal{O}(\ln(K) \ln(T))$).

The work Pasteris et al. (2023) then considered the problem in which, instead of the contexts being vertices of a known graph, they instead come from an arbitrary metric space of finite doubling dimension, and are not known to Learner a-priori. They noted that a tree can be formed by connecting each context to an approximate nearest neighbour in the set of all contexts seen previously: which can be found efficiently online with data-structures such as *navigating nets* Krauthgamer & Lee (2004). If this tree was known a-priori then either of the GABA algorithms could be used. However, since the tree is not known a-priori, the GABA algorithms fail to work. Hence, Pasteris et al. (2023) introduced the CBNN algorithm which runs in a per-trial time of $\tilde{\mathcal{O}}(\ln(K) \ln(T)^2)$ and works with any tree constructed in an online fashion. Although CBNN, which utilises the rebalancing ternary tree of Matsuzaki & Morihata (2008) and the online belief propagation algorithm of Delcher et al. (1995), borrows elements from both of the GABA algorithms it is radically different, introducing several new data-structures. Pasteris et al. (2023) analysed the case in which the tree is constructed by the approximate nearest neighbour method given above. In this paper we utilise CBNN but construct the tree in a different way.

## 1.2 Notation

We define $\mathbb{N}$ to be the set of natural numbers, not including 0. Given $N \in \mathbb{N}$ we define $[N] := \{i \in \mathbb{N} \mid i \leq N\}$. Given a predicate $P$ we define $[\![P]\!] := 0$ if $P$ is false and define $[\![P]\!] := 1$ if $P$ is true. Please see Appendix A for a table of the notation that will be introduced in this paper.

## 2 Problem Description

We consider the following game between *Nature* (our adversary) and *Learner* (our algorithm). We have $T$ *trials* and $K$ *actions*. Nature first chooses a matrix $\mathbf{\Delta} \in [0,1]^{T \times T}$ satisfying the following conditions:

- For all $s, t \in [T]$ we have $\Delta_{s,t} = \Delta_{t,s}$.

- For all $t \in [T]$ we have $\Delta_{t,t} = 0$.

- For all $r, s, t \in [T]$ we have $\Delta_{r,t} \leq \Delta_{r,s} + \Delta_{s,t}$. This property is called the *triangle inequality*.

For all $s, t \in [T]$ we call $\Delta_{s,t}$ the *distance* between trials $s$ and $t$. This distance represents the similarity between trials $s$ and $t$: a smaller distance indicating a greater degree of similarity. Intuitively, we implicitly have some metric space and every trial $t \in [T]$ is implicitly associated with a *context* $x_t$ in the metric space. $\Delta_{s,t}$ is then the distance from $x_s$ to $x_t$ in the metric space.

For all trials $t \in [T]$ and actions $a \in [K]$, Nature chooses a probability distribution $\tilde{\ell}_{t,a}$ over $[0,1]$ and a *loss* $\ell_{t,a}$ is then drawn from $\tilde{\ell}_{t,a}$. We note that Learner has knowledge of only $T$ and $K$ (although the requirement of knowledge of $T$ can be removed by a simple doubling trick).

The game then proceeds in $T$ trials. On trial $t$ the following happens:

1. For all $s \in [t]$ the distance $\Delta_{s,t}$ is revealed to Learner.

2. Learner stochastically chooses an action $a_t \in [K]$.

3. The loss $\ell_{t,a_t}$ is revealed to Learner.

Our measurement of Learner's performance is by the following notion of *regret*. A *policy* is any vector $\boldsymbol{y} \in [K]^T$ in which for all $s, t \in [T]$ with $\Delta_{s,t} = 0$ we have $y_s = y_t$. That is: a policy $\boldsymbol{y}$ associates each trial $t$ with an action $y_t$, where trials with identical implicit contexts are associated with identical actions (as they are indistingishable). Given such a policy $\boldsymbol{y}$, we define the *regret* of Learner, with respect to $\boldsymbol{y}$, as:

$$R(\boldsymbol{y}) := \sum_{t \in [T]} \mathbb{E}[\ell_{t,a_t} - \ell_{t,y_t}]$$

which is the expected difference between the cumulative loss of Learner and that which would have been obtained by following policy $\boldsymbol{y}$. A policy $\boldsymbol{y}$ is (informally) considered *natural* if typically $y_s = y_t$ when $\Delta_{s,t}$ is small. The *inductive bias* is towards more natural policies: i.e. a more natural policy $\boldsymbol{y}$ should yield a smaller regret $R(\boldsymbol{y})$.

Note that, since Nature has complete control over each distribution $\tilde{\ell}_{t,a}$, our problem generalises the fully adversarial problem (which is the special case in which each distribution $\tilde{\ell}_{t,a}$ is a delta function). We are considering this generalised problem in order for our bound to be better when there is an element of stochasticity in Nature's choices.

## 3 The Algorithm

We now describe our algorithm HNN. The algorithm takes parameters $c \geq 1$, $\epsilon \geq 0$ and $\rho > 0$. Our performance is best when $c = 1$ and $\epsilon = 0$. However, we allow these parameters to take on higher values in order to make our algorithm more efficient. Specifically, whenever $c > 1$ and $\epsilon$ is polynomial in $1/T$ our per trial time complexity is $\mathcal{O}(K \ln(T)^2)$ when the dimensionality is bounded. We will briefly describe how one can use the methodology Pasteris et al. (2023) to speed up the algorithm to $\mathcal{O}(\ln(K) \ln(T)^2)$ time per trial, referring the interested reader to that paper.

### 3.1 The Tree

HNN works by maintaining a growing rooted tree $\mathcal{T}$ where, at the start of each trial $t > 1$, $\mathcal{T}$ contains, as its vertices, the set $[t-1]$. The root of the tree is equal to 1, which is its single vertex on trial 1. Given any $t \in [T]$, once $t$ has been added to $\mathcal{T}$ we define $p_t$ to be the parent of $t$. The difference between HNN and NN is the way in which $\mathcal{T}$ is constructed - whereas NN chooses $p_t$ to be an approximate nearest neighbour of $t$ in $[t-1]$, HNN utilises a more complex method (inspired by Slivkins (2009)) to find $p_t$.

In order to grow $\mathcal{T}$ we will maintain a number $h \in \mathbb{N} \cup \{0\}$ initialised equal to 0 as well as, for all $d \in [h] \cup \{0\}$, a set $\mathcal{H}_d \subseteq [T]$ initialised so that $\mathcal{H}_0 = 1$. We now describe how the tree $\mathcal{T}$ is updated at the start of each trial $t > 1$. To do this we first define, for any non-empty set $\mathcal{H} \subseteq [T]$ and any trial $t \in [T]$, a *c-nearest neighbour* of $t$ in $\mathcal{H}$ to be any trial $s \in \mathcal{H}$ in which:

$$\Delta_{s,t} \leq c \min\{\Delta_{r,t} \mid r \in \mathcal{H}\}.$$

At the start of each trial $t > 1$ HNN does the following:

---

**Algorithm: First part of trial $t$.**

1. For all $d \in [h] \cup \{0\}$ let $s_d$ be a $c$-nearest neighbour of $t$ in $\mathcal{H}_d$.

2. If there exists $d \in [h] \cup \{0\}$ with $\Delta_{s_d,t} \leq \epsilon$ then $p_t \leftarrow s_d$ for any such $d$.

3. If $\Delta_{s_d,t} > \epsilon$ for all $d \in [h] \cup \{0\}$ then:

    (a) $\delta \leftarrow \max\{d \in [h] \,|\, \Delta_{s_d,t} \leq 1/2^d\}$.
    (b) If $\delta = h$ then:
        i. $h \leftarrow h + 1$.
        ii. $\mathcal{H}_h \leftarrow \emptyset$.
    (c) $\mathcal{H}_{\delta+1} \leftarrow \mathcal{H}_{\delta+1} \cup \{t\}$.
    (d) $p_t \leftarrow s_\delta$.

---

We note that step 2 of this subroutine is a *binning* step: if $\Delta_{s_d,t} \leq \epsilon$ for some $d$ then we treat trial $t$ as if it had an implicit context identical to that of trial $s_d$. However, care must be taken so that the tree $\mathcal{T}$ does not become too deep. This is achieved by setting $p_t := s_d$ and then excluding $t$ from the sets $\mathcal{H}_d$. It is crucial that $t$ is not added to any such set since we have an identical trial $s_d$ in the data-structure already. We note that the only effect that this binning will have on our bounds is by moving the implicit contexts by a distance of up to $\epsilon$. The purpose of binning is it that it bounds the minimum distance between trials in the data-structure by $\epsilon/c$: which is required for the tree $\mathcal{T}$ to be not too deep and also required in order to find the approximate nearest neighbours efficiently, as we now describe.

Given that $c > 1$, $\epsilon$ is polynomial in $1/T$, and the metric $\Delta$ has bounded doubling dimension, we can run the above subroutine in time $\mathcal{O}(\ln(T)^2)$ by maintaining the *navigating net* of Krauthgamer & Lee (2004) over each set $\mathcal{H}_d$. Specifically, since the aspect ratio (ratio between the maximum and minimum inter-trial distances) of $\mathcal{H}_d$ is always at least $\epsilon/c$, and hence polynomial in $1/T$, the navigating net allows us to find the $c$-nearest neighbour of $t$ in $\mathcal{H}_d$ in time $\mathcal{O}(\ln(T))$. By the same argument, the navigating net can also update in a time of $\mathcal{O}(\ln(T))$ whenever a trial is added to $\mathcal{H}_d$. Since, due to the binning step, we always have $h \in \mathcal{O}(\ln(T))$ we obtain the per-trial time complexity of $\mathcal{O}(\ln(T)^2)$.

### 3.2 Belief Propagation

On any trial $t$, once $t$ has been added to the tree $\mathcal{T}$ we select $a_t$ and then incorporate the loss $\ell_{t,a_t}$ via the *online belief propagation* algorithm of Delcher et al. (1995) but incorporating the fact that $\mathcal{T}$ is growing. To do this we maintain, for each $s \in [T]$, a dynamic (in that it changes from trial to trial) vector $\boldsymbol{e}_s \in \mathbb{R}^K$ initialised so that $e_{s,a} = 1$ for all $a \in [K]$. For all $a, b \in [K]$ we define the quantity:

$$\tau_{a,b} := [\![a = b]\!] \left(1 - \frac{1}{T}\right) + [\![a \neq b]\!] \frac{1}{(K-1)T}.$$

We note that for any vector $\boldsymbol{v} \in \mathbb{R}^K$ we can compute the vector $\boldsymbol{v}' \in \mathbb{R}^K$ defined by $v'_b := \sum_{a \in [K]} v_a \tau_{a,b}$ in time $\mathcal{O}(K)$. Specifically, we have:

$$v'_b = \left(1 - \frac{1}{T}\right) v_b + \frac{\left(\sum_{a \in [K]} v_a\right) - v_b}{(K-1)T}$$

On any trial $t \in [T]$, once $t$ has been added to the tree $\mathcal{T}$ we do as follows.

---

**Algorithm: Second part of trial $t$ (when using belief propagation).**

1. For all $a \in [K]$ set:
$$\varphi_{1,a} \leftarrow 1 \,.$$

2. Descend the tree $\mathcal{T}$ from 1 (i.e. the root) to $t$. When at a vertex $s \neq 1$ set, for all $a \in [K]$, the quantities:
$$\vartheta_{s,a} \leftarrow \sum_{b \in [K]} e_{s,b} \tau_{b,a} \quad ; \quad \varphi_{s,a} \leftarrow \sum_{b \in [K]} \left( \frac{\varphi_{p_s,b} e_{p_s,b}}{\vartheta_{s,b}} \right) \tau_{b,a} \,.$$

3. For all $a \in [K]$ set:
$$\pi_a \leftarrow \frac{\varphi_{t,a}}{\sum_{b \in [K]} \varphi_{t,b}} \,.$$

4. Sample $a_t$ such that for all $a \in [K]$ we have $a_t = a$ with probability $\pi_a$.

5. Play $a_t$.

6. Receive the loss $\ell_{t,a_t}$.

7. For all $a \in [K]$ set:
$$e_{t,a} \leftarrow \exp\left( \frac{-[\![a_t = a]\!] \ell_{t,a_t}}{\rho \pi_a \sqrt{KT}} \right) \,.$$

8. Ascend the tree $\mathcal{T}$ from $t$ to 1 (i.e. the root). When at a vertex $s \neq 1$ set, for all $a \in [K]$, the quantities:
$$\vartheta'_{s,a} \leftarrow \sum_{b \in [K]} e_{s,b} \tau_{b,a} \quad ; \quad e_{p_s,a} \leftarrow \frac{\vartheta'_{s,a} e_{p_s,a}}{\vartheta_{s,a}} \,.$$

---

To aid the reader, we now give a brief verbal overview of online belief propagation. As explained above, we maintain a vector $\boldsymbol{e}_s \in \mathbb{R}^K$ for each vertex/trial $s$. These vectors are initialised such that each component is equal to 1. For each trial $t$ there are two phases: the first phase selects the action $a_t$ and the second phase updates some of the vectors in $\{\boldsymbol{e}_s \,|\, s \in [t]\}$ after receiving the loss $\ell_{t,a_t}$. The first phase proceeds by descending the tree along the path from vertex/trial 1 to vertex/trial $t$. For each vertex/trial $s$ encountered during the descent, a vector $\boldsymbol{\varphi}_s \in \mathbb{R}^K$ is computed. When $s = 1$ (i.e. $s$ is the root) $\boldsymbol{\varphi}_s$ is defined to be the vector with all components equal to 1, and when $s \neq 1$ (so that $s$ has a parent $p_s$ and $\boldsymbol{\varphi}_{p_s}$ has already been computed) $\boldsymbol{\varphi}_s$ is computed from $\boldsymbol{\varphi}_{p_s}$, $\boldsymbol{e}_s$ and $\boldsymbol{e}_{p_s}$ (see Line 2). Once $\boldsymbol{\varphi}_t$ has been constructed, it is normalised to form a probability vector $\boldsymbol{\pi}$ from which $a_t$ is then drawn. In the second phase, $\ell_{t,a_t}$ is first received and used to update $\boldsymbol{e}_t$ (see Line 7). The tree is then climbed up the path from $t$ to 1. When at a non-root vertex/trial $s$ on this path, $\boldsymbol{e}_{p_s}$ is updated based on its old value and the new and old values of $\boldsymbol{e}_s$ (see Line 8, noting that $\boldsymbol{\vartheta}_s$ was constructed in Line 2).

Note that running online belief propagation on any trial takes a time of $\mathcal{O}(hK)$ which is $\mathcal{O}(K \ln(T))$ under the above conditions.

### 3.3 CBNN

In order to achieve, under the above conditions, a per-trial time complexity of $\mathcal{O}(\ln(K) \ln(T)^2)$, we can utilise the CBNN algorithm of Pasteris et al. (2023) instead of belief propagation. CBNN solves the following problem. A-priori Nature chooses, for each trial $t \in [T]$, a loss vector $\boldsymbol{\ell}_t \in [0,1]^K$ and, when $t > 1$, a trial $n(t) \in [t-1]$. The idea is that the trial $n(t)$ is in some way *similar* to the trial $t$. On each trial $t \in [T]$ the following happens:

1. If $t > 1$ then $n(t)$ is revealed to Learner.

2. Learner chooses an action $a_t \in [K]$.

3. The loss $\ell_{t,a_t}$ is revealed to Learner.

CBNN is an algorithm for Learner in this problem. It takes a parameter $\rho > 0$ and, given any policy $\boldsymbol{y} \in [K]^T$, achieves:

$$\sum_{t \in [T]} \mathbb{E}[\ell_{t,a_t} - \ell_{t,y_t}] \in \mathcal{O}\left( \left( \rho + \frac{\Phi(\boldsymbol{y})}{\rho} \right) \sqrt{KT} \right)$$

where:

$$\Phi(\boldsymbol{y}) := 1 + \sum_{t \in [T] \setminus \{1\}} [\![ y_t \neq y_{n(t)} ]\!] .$$

The time complexity of CBNN is only $\mathcal{O}(\ln(K) \ln(T)^2)$ per trial. We refer the reader to Pasteris et al. (2023) for a complete description of the mechanics of CBNN.

HNN utilises CBNN by setting $n(t) := p_t$ for all $t \in [T] \setminus \{1\}$. The losses revealed to CBNN are those received by HNN. The parameter $\rho$ of CBNN is identical to that of HNN. The actions selected by HNN are those selected by CBNN.

We note that CBNN works on what it calls *ternary search trees* to improve efficiency when the tree is deep. However, in HNN the tree $\mathcal{T}$ has a depth of only $\mathcal{O}(\ln(c/\epsilon))$ so we don't need to use ternary search trees, instead maintaining *contractions* (as described in Pasteris et al. (2023)) over the original tree instead.

### 3.4  Computational Complexity

The following theorem states the time complexities of the various versions of HNN.

**Theorem 3.1.** *For metric spaces of bounded dimensionality and with $\epsilon$ polynomial in $1/T$ and $c > 1$, the per-trial time complexities of HNN with belief propagation and CBNN are $\mathcal{O}(\ln(T)^2 + K \ln(T))$ and $\mathcal{O}(\ln(K) \ln(T)^2)$ respectively. For general metric spaces the per-trial time complexities of HNN with belief propagation and CBNN are $\mathcal{O}(KT)$ and $\tilde{\mathcal{O}}(T)$ respectively.*

## 4  Performance in Euclidean Space

Before we give our general regret bound we will first give, as an illustration, a corollary of it for the important special case in which our implicit contexts lie in a euclidean space. We call this special case the *euclidean bandit problem*. We also give an almost matching lower bound, highlight our improvement over NN, and give a bound for the i.i.d. stochastic special case. We note that our more general bound in Section 5 improves on the bound of this section, as well as generalising it to arbitrary metrics.

### 4.1  The Euclidean Bandit Problem

In the euclidean bandit problem there exists some constant $d$ which is the dimensionality of the dataset. For all $x \in \mathbb{R}^d$ let $\|x\|$ be the euclidean norm of $x$. Given $x \in \mathbb{R}^d$ and $r > 0$ we define the *ball* with *centre* $x$ and *radius* $r$ as:

$$\mathcal{B}(x,r) := \{x' \in \mathbb{R}^d \,|\, \|x - x'\| \leq r\} .$$

We define $\mathcal{X} := \mathcal{B}(0, 1/2)$. Nature chooses, a-priori, a sequence of *contexts*:

$$\langle x_t \,|\, t \in [T] \rangle \subseteq \mathcal{X}$$

unknown to Learner. The matrix $\boldsymbol{\Delta}$ is then defined such that for all $s, t \in [T]$ we have:

$$\Delta_{s,t} := \|x_s - x_t\|$$

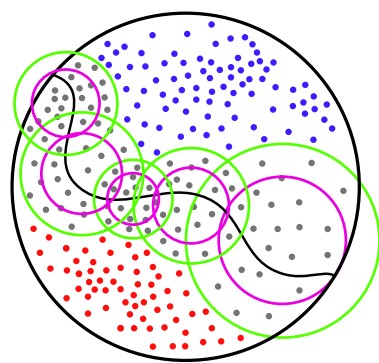

Figure 1: An example with $K = 2$ and $d = 2$. Here we have $N = 5$. The black curve is a decision boundary of $\boldsymbol{y}$. The purple balls are those in the boundary cover $\{\mathcal{B}(v_i, r_i) \,|\, i \in [N]\}$ and the green balls are those in $\{\mathcal{B}(v_i, \xi r_i) \,|\, i \in [N]\}$. The grey contexts are those in $\{x_t \,|\, t \in \mathcal{M}\}$, the red contexts are those in $\{x_t \,|\, t \in [T] \setminus \mathcal{M} \wedge y_t = 1\}$, and the blue contexts are those in $\{x_t \,|\, t \in [T] \setminus \mathcal{M} \wedge y_t = 2\}$. Note that the purple balls cover the decision boundary and the grey contexts are those covered by the green balls.

## 4.2 Regret

Our regret bound for the euclidean bandit problem is based on the following concept of a *boundary cover* which is (informally) any set of balls covering the decision boundary of the comparator policy. Formally, take some arbitrary policy $\boldsymbol{y} \in [K]^T$. An *extension* of $\boldsymbol{y}$ is any function $\tilde{y} : \mathcal{X} \to [K]$ such that $\tilde{y}(x_t) = y_t$ for all $t \in [T]$. A *decision boundary* of $\boldsymbol{y}$ is defined as a set:

$$\{x \in \mathcal{X} \,|\, \forall r > 0 \,,\, \exists x' \in \mathcal{B}(x, r) \cap \mathcal{X} : \tilde{y}(x') \neq \tilde{y}(x)\}$$

where $\tilde{y}$ is some arbitrary extension of $\boldsymbol{y}$. Finally, we define a *boundary cover* of $\boldsymbol{y}$ as any set of balls which covers a decision boundary of $\boldsymbol{y}$.

With the definition of a boundary cover in hand, we have the following bound on the regret of HNN.

**Theorem 4.1.** *Let $C > 0$ and $\xi > 1$ be arbitrary constants and assume HNN is run on the euclidean bandit problem with any $c \geq 1$, any $\epsilon \leq (\xi - 1)T^{-C}/2$ and any $\rho > 0$. Then given any policy $\boldsymbol{y} \in [K]^T$ and any boundary cover:*

$$\{\mathcal{B}(v_i, r_i) \,|\, i \in [N]\}$$

*of $\boldsymbol{y}$ with:*

$$\min\{r_i \,|\, i \in [N]\} \geq T^{-C}$$

*we have:*

$$R(\boldsymbol{y}) \leq |\mathcal{M}| + \tilde{\mathcal{O}}\left(\left(\rho + \frac{N}{\rho}\right)\sqrt{KT}\right)$$

*where:*

$$\mathcal{M} := \left\{t \in [T] \,\middle|\, x_t \in \bigcup_{i \in [N]} \mathcal{B}(v_i, \xi r_i)\right\}$$

*Proof.* A corollary of Theorem 5.1. See Appendix D. $\qquad\square$

Theorem 4.1 is illustrated in Figure 1. Note that $N$ is the number of balls in the boundary cover and $|\mathcal{M}|$ is the number of contexts lying inside the balls of the boundary cover when each is expanded by a factor $\xi$.

We note that Theorem 4.1 is a corollary of Theorem 5.1. However, using Theorem 5.2 instead can significantly reduce the term $|\mathcal{M}|$ that appears in the regret. This improvement allows us to exploit Holder continuity in the i.i.d. stochastic bound coming up in Section 4.5.

### 4.3 Lower Bound

We note that we have the following regret lower bound which is extremely close to our upper bound in Theorem 4.1 when $\rho = \sqrt{N}$.

**Theorem 4.2.** *For any algorithm, given any $\xi > 1$ and $r > 0$ with $8\xi r < 1$, and any $N' \in \mathbb{N}$ less than the $2\xi r$-packing number of $\mathcal{X}$ and any $M \leq T/2$, then if $T \geq 2N'K$ there exists:*

- *A policy $\boldsymbol{y} \in [K]^T$.*

- *A boundary cover of $\boldsymbol{y}$ of cardinality $N \in \Theta(N')$ in which each ball has radius $r$.*

- *A choice of contexts and losses by Nature such that, when $\mathcal{M}$ is defined from $\xi$ and the boundary cover as in Theorem 4.1, we have $|\mathcal{M}| = M$.*

*such that:*

$$R(\boldsymbol{y}) \geq |\mathcal{M}|/2 + \Omega(\sqrt{NKT})$$

*Proof.* See Appendix E □

We also note that for any algorithm we require, for Theorem 4.1 to hold, the condition that $\xi > 1$. Specifically, we have the following theorem:

**Theorem 4.3.** *Given any algorithm there exists a policy $\boldsymbol{y}$, a boundary cover of $\boldsymbol{y}$ with cardinality 2, and a choice of contexts and losses by Nature, in which no context lies in either ball of the boundary cover and $R(\boldsymbol{y}) \geq T/2$.*

*Proof.* See Appendix F □

### 4.4 Comparison to Nearest Neighbour

We now compare the regret bound of HNN in euclidean space, as given in Theorem 4.1, to that of NN, which is, as far as we are aware, the only other algorithm that can handle the fully adversarial problem. We assume here that NN is using the optimal binning radius (which it must know a-priori).

The regret bound of NN is like that in Theorem 4.1 except with the restriction that all the balls in the boundary cover must have the same radii (a constant multiple of the binning radius). This is essentially proved in the proof of Theorem 3.9 of Pasteris et al. (2023). The restriction of all balls having the same radii severely limits NN: meaning that, unlike HNN, it cannot adapt to the context density and decision boundary smoothness. We now give an illustrative example of its inability to adapt to context density.

For simplicity let's assume that the parameter $\rho$, of NN and HNN, is set equal to a constant, although the argument easily extends to tuned parameter values as well. Consider two disjoint balls $\mathcal{B}, \mathcal{B}'$ in $\mathbb{R}^2$ with radii equal to 1 and $r < 1$ respectively (noting that the diameter of the dataset in our problem can actually be any constant). Assume that each ball has $T/2$ contexts distributed uniformly over it. Suppose we have two actions and a comparator policy such that the decision boundary (of the policy) on each ball is a straight line going through the ball's centre. This is depicted in Figure 2.

First let's analyse NN when working on each ball as a seperate problem. Given a binning radius of $\varphi$, the regret of NN on $\mathcal{B}$ is $\tilde{\mathcal{O}}(\sqrt{T}/\varphi + \varphi T)$ and the regret of NN on $\mathcal{B}'$ is $\tilde{\mathcal{O}}(r\sqrt{T}/\varphi + \varphi T/r)$. For both these problems the optimal value of $\varphi$ leads to a regret of $\tilde{\mathcal{O}}(T^{3/4})$.

However, things change when NN is working on both balls at the same time. In this case, given a binning radius of $\varphi$, the regret of NN is:

$$\tilde{\mathcal{O}}((\sqrt{T}/\varphi + \varphi T) + (r\sqrt{T}/\varphi + \varphi T/r)) = \tilde{\mathcal{O}}(\sqrt{T}/\varphi + \varphi T/r)$$

i.e. the sum of the regrets on both balls. This means that the optimal value of $\varphi$ leads to a regret of $\tilde{\mathcal{O}}(T^{3/4}r^{-1/2})$ which can be dramatically higher than if the balls were learnt separately. In fact, when

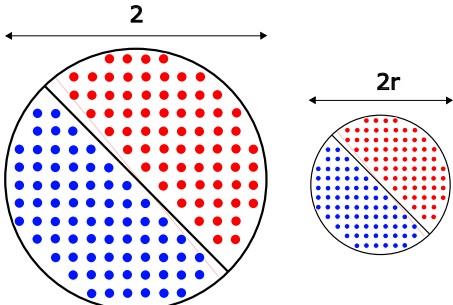

Figure 2: An example of when NN performs poorly. The colour of a context represents the action assigned to it by the comparator policy.

$r \leq T^{-1/2}$ this bound is vacuous. Our algorithm HNN, however, has a regret of $\tilde{\mathcal{O}}(T^{3/4})$ whenever $1/r$ is polynomial in $T$ - the same as if the balls were learnt seperately.

When working in $\mathbb{R}^D$ for $D > 2$, the improvement of HNN over NN is even stronger. To see this note that the regret of NN when working on both balls is:

$$\tilde{\mathcal{O}}((\sqrt{T}/\varphi^{D-1} + \varphi T) + (\sqrt{T}(r/\varphi)^{D-1} + \varphi T/r)) = \tilde{\mathcal{O}}(\sqrt{T}/\varphi^{D-1} + \varphi T/r)$$

so that the optimal value of $\varphi$ gives us a regret of $\tilde{\mathcal{O}}\left(r^{-(D-1)/D} T^{(2D-1)/(2D)}\right)$ which becomes vacuous at $r \leq T^{-1/(2D-2)}$. HNN, on the other hand, achieves a regret of $\tilde{\mathcal{O}}(T^{(2D-1)/(2D)})$ whenever $1/r$ is polynomial in $T$ - the same as if the balls were learnt seperately. Note that $\rho$ is set to a constant here: when, instead, $\rho = T^{(D-1)/2(D+1)}$ then HNN achieves a regret of $\tilde{\mathcal{O}}(T^{D/(D+1)})$.

### 4.5 Stochastic Bandits

Even though HNN is designed for the adversarial problem, we will now show that that in the i.i.d. stochastic special case it is competitive against algorithms designed purely for that special case, sometimes even outperforming. In fact, our experimental results in Section 6 show that HNN can outperform such algorithms in real world i.i.d. stochastic problems. For HNN the bound we give is crude, and we believe it can also be obtained by NN (with the optimal binning radius). Our bound improves on the stochastic bound of NN given in Pasteris et al. (2023) by allowing for the Bayes decision boundary to have fractal dimension greater than $d - 1$ and by becoming adaptive to Holder continuous mean losses.

We maintain the notation and problem description given at the start of this section. In the i.i.d. stochastic special case we have an unknown probability density $\nu$ over the space $\mathcal{X} \times [0, 1]^K$ and for every trial $t \in [T]$ the pair $(x_t, \ell_t)$ is drawn from $\nu$. We assume that $\nu$ is bounded above by a constant. Given $x \in \mathcal{X}$ and $a \in [K]$ we define:

$$\bar{\nu}_a(x) := \mathbb{E}_{(x', \ell') \sim \nu}[\ell'_a \mid x' = x]$$

which is the mean loss of action $a$ when given context $x$. We define the *Bayes optimal classifier* $\tilde{y} : \mathcal{X} \to [K]$ such that for all $x \in \mathcal{X}$ we have:

$$\tilde{y}(x) := \mathrm{argmin}_{a \in [K]} \bar{\nu}_a(x)$$

and define the *Bayes decision boundary* $\mathcal{D}$ as the set:

$$\{x \in \mathcal{X} \mid \forall r > 0, \, \exists x' \in \mathcal{B}(x, r) \cap \mathcal{X} : \tilde{y}(x') \neq \tilde{y}(x)\}$$

Furthermore, we define the *decision boundary dimension* $\vartheta$ such that the exists $c' > 0$ such that for all $r > 0$ we have that $\mathcal{D}$ can be covered by $c' r^{-\vartheta}$ balls of radius $r$. Finally we define the *Holder complexity* $\varphi$ such that there exists $\hat{c} > 0$ such that for all $x, x' \in \mathcal{X}$ we have:

$$\max_{a \in [K]} |\bar{\nu}_a(x) - \bar{\nu}_a(x')| \leq \hat{c} \|x - x'\|^\varphi$$

We now state the performance of HNN.

**Theorem 4.4.** *When HNN is run with $\epsilon \leq 1/T$ and the optimal value of $\rho$ in the i.i.d. stochastic bandit problem as described above then given the policy $\boldsymbol{y}$ is such that $y_t = \tilde{y}(x_t)$ for all $t \in [T]$, we have:*

$$\mathbb{E}\left[R(\boldsymbol{y})\right] \in \tilde{\mathcal{O}}\left(T(K/T)^{\frac{\varphi+d-\vartheta}{2\varphi+2d-\vartheta}}\right)$$

*where the $\tilde{\mathcal{O}}$ suppresses a constant dependent on $\nu$.*

*Proof.* A corollary of Theorem 5.2. See Appendix G. □

We note that this bound is crude and does not account for the ability of HNN to adapt to varying density and varying values of $\varphi$ and $\vartheta$ (along with their corresponding constants $c'$ and $\hat{c}$) across the space. However, we believe it to be novel, in some cases improving over bounds attained by algorithms designed specifically for the i.i.d. stochastic problem. In particular, we now compare it to the bound of the ABSE algorithm of Perchet & Rigollet (2011) which is, as far as we are aware, the state of the art for i.i.d. stochastic bandits in euclidean space. Unlike our bound, the bound of ABSE depends upon a quantity $\alpha > 0$ defined as follows. First define, for all $x \in \mathcal{X}$, the *gap* as:

$$g(x) := \min_{a \in [K]\,:\,\bar{\nu}_a(x) \neq \bar{\nu}_{\tilde{y}(x)}(x)} \left(\bar{\nu}_a(x) - \bar{\nu}_{\tilde{y}(x)}(x)\right)$$

and then let $\alpha$ be equal to the minimum value of $\alpha'$ such that there exists $c^\dagger > 0$ such that for all $\delta > 0$ we have:

$$\mathbb{P}_{(x,\boldsymbol{\ell}')\sim\nu}[g(x) \leq \delta] \leq c^\dagger \delta^{\alpha'}$$

Given $\alpha > 0$, the bound obtained by ABSE is then:

$$\mathbb{E}\left[R(\boldsymbol{y})\right] \in \tilde{\mathcal{O}}\left(T(K/T)^{\frac{\varphi(1+\alpha)}{2\varphi+d}}\right)$$

Note that this is, in general, incomparable to our bound in Theorem 4.4. For example, as $\varphi$ approaches $0$ whilst $\vartheta < d$ and $\alpha$ stay constant, or when $\alpha$ approaches $0$ whilst $\vartheta < d$ and $\varphi$ stay constant, our bound is an improvement.

## 5 Performance in General Metric Spaces

We now bound the expected regret of HNN for general matrices $\boldsymbol{\Delta}$. The final bound also improves on the bound in Theorem 4.1 when in euclidean space. We will assume here that the parameter $\epsilon$ is set equal to $0$. This is without loss of generality as for $\epsilon > 0$ the result is the same but such that distances have been shifted by up to $2\epsilon$ (which is typically negligible). In what follows, given $c$, we have universal constants $\lambda > 1$ and $z > 0$.

Theorem 4.1 involved a set of trials $\mathcal{M}$ whose contexts lay near the decision boundary of the comparator policy. In general, a *margin* is any set $\mathcal{M} \subset [T]$ in which for all $s \in \mathcal{M}$ and all $t \in [T]$ with $\Delta_{s,t} = 0$ we have $t \in \mathcal{M}$. Our general bound will hold for any possible margin $\mathcal{M} \subseteq [T]$. So suppose we have any arbitrary policy $\boldsymbol{y} \in [K]^T$ and margin $\mathcal{M} \subseteq [T]$. Note that the algorithm has no knowledge of either $\boldsymbol{y}$ or $\mathcal{M}$.

### 5.1 Packing Complexity

Here we give our first, simpler, general regret bound. For all $t \in [T]$ we define:

$$\gamma_t := \min\{\Delta_{s,t} \,|\, s \in [T] \setminus \mathcal{M} \wedge y_s \neq y_t\}$$

which is the minimum distance from $t$ to any trial $s \notin \mathcal{M}$ with a differing label. We also define:

$$\Lambda := \min\{\gamma_t \,|\, t \in [T]\}\,.$$

which can be viewed as the *width* of the margin.

We define the *packing complexity* $\Psi$ as the maximum cardinality of any set $\mathcal{S} \subseteq [T]$ in which for all $s, t \in \mathcal{S}$ with $s \neq t$ we have:

$$\Delta_{s,t} > z\gamma_t .$$

Note that the packing complexity is the maximum cardinality of any packing of balls such that, if a ball's centre is $t$, then its radius is $z\gamma_t$. Here we define a packing to mean that the centre of any ball is not contained inside any other ball.

With the definition of packing complexity in hand, we now present our first general regret bound.

**Theorem 5.1.** *Given any policy $\boldsymbol{y} \in [K]^T$ and any margin $\mathcal{M} \subseteq [T]$, then if $\Psi$ and $\Lambda$ are defined as above we have that HNN with $\epsilon = 0$ has an expected regret of:*

$$R(\boldsymbol{y}) \leq |\mathcal{M}| + \tilde{\mathcal{O}}\left(\left(\rho + \frac{\Psi \ln(1/\Lambda)^2}{\rho}\right)\sqrt{KT}\right) .$$

*Proof.* An immediate corollary of Theorem 5.2. □

When $\mathcal{M} = \emptyset$ (and the aspect ratio is polynomial in $T$) our bound is similar to that of Pasteris et al. (2023) (without binning) except that their equivalent of $\Psi$ is a covering number rather than a packing number (the radii of the balls in the covering/packing being a constant factor different in the two bounds). However, we have a big advantage over Pasteris et al. (2023) in that we can choose any margin $\mathcal{M} \subseteq [T]$ which we can essentially hold out in the computation of the values $\gamma_t$. As the margin increases the values $\gamma_t$ increase, meaning that $\Psi$ decreases. In return for the decrease in $\Psi$, the factor $|\mathcal{M}|$ is added to the regret. There is a sweet-spot: the optimal margin. The purpose of our margin is essentially the same purpose as using binning in Pasteris et al. (2023). However, since our margin can be any set we have much more flexibility (compared to the constant binning radius of Pasteris et al. (2023)), allowing us to adapt locally to the density of contexts and smoothness of decision boundary: as was illustrated in Section 4.4.

## 5.2 Excesses

We will now tighten the regret bound in Theorem 5.1 by reducing the term $|\mathcal{M}|$, which is what allows us to exploit Holder continuity in Theorem 4.4. To do this we will now define, for all $t \in \mathcal{M}$, a quantity $\mu_t$ that we call the *excess* of trial $t$. First define:

$$\theta_t := \min\{\Delta_{s,t} \,|\, s \in [T] \setminus \mathcal{M}\}$$

which is the minimum distance from $t$ to any trial not in the margin. Define:

$$\mathcal{Y}_t := \{y_s \,|\, s \in [T] \,\wedge\, \Delta_{s,t} \leq \lambda\theta_t\}$$

which is the set of labels of trials within a distance of $\lambda\theta_t$ from $t$. Finally define:

$$\mu_t := \max\{\mathbb{E}[\ell_{t,a} - \ell_{t,y_t}] \,|\, a \in \mathcal{Y}_t\}$$

which is the difference between maximum expected loss, on trial $t$, of any action in $\mathcal{Y}_t$, and the expected loss, on trial $t$, of the action $y_t$.

We have now defined all the quantities needed to present our second general regret bound:

**Theorem 5.2.** *Given any policy $\boldsymbol{y} \in [K]^T$ and any margin $\mathcal{M} \subseteq [T]$, then if $\Psi, \Lambda$ and $\{\mu_t \,|\, t \in \mathcal{M}\}$ are defined as above then HNN with $\epsilon = 0$ has an expected regret of:*

$$R(\boldsymbol{y}) \leq \sum_{t \in \mathcal{M}} \mu_t + \tilde{\mathcal{O}}\left(\left(\rho + \frac{\Psi \ln(1/\Lambda)^2}{\rho}\right)\sqrt{KT}\right) .$$

*Proof.* See Appendix C. □

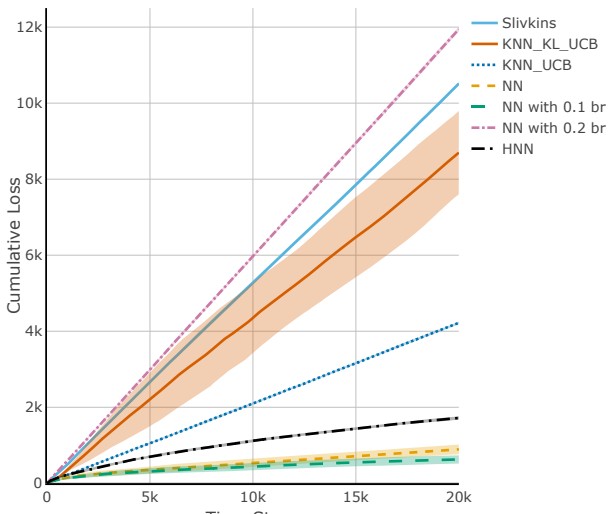
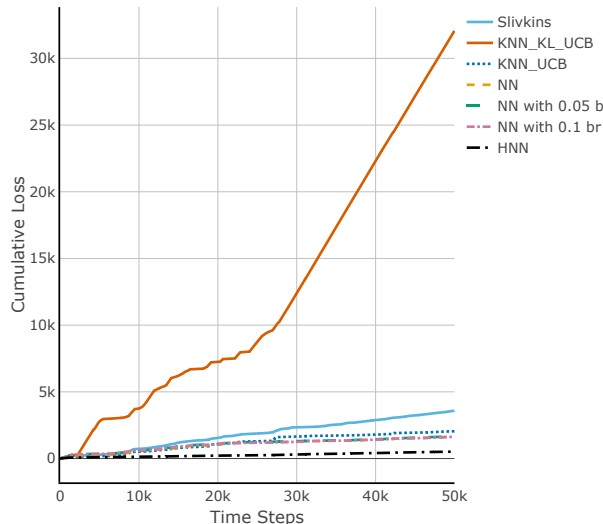

(a) Cumulative loss averaged over 10 runs for the UCI firewall dataset. The lines represent the average over 10 independent runs. Shaded regions indicate $\pm 1$ standard deviation. For most methods, the standard deviation is very small and, as such, not visually obvious in the plot.

(b) Cumulative loss for the CICIDS2017 intrusion dataset experiment.

Figure 3: Comparison of cumulative loss across two real-world datasets.

To compare against Theorem 5.1 we note that $|\mathcal{M}|$ has been replaced by $\sum_{t \in \mathcal{M}} \mu_t$. Typically $\sum_{t \in \mathcal{M}} \mu_t$ is much less than $|\mathcal{M}|$ since, for any trial $t \in \mathcal{M}$, any action $a \in \mathcal{Y}_t$ is typically equal to $y_s$ for some $s$ close to $t$. If $\boldsymbol{y}$ is a good comparator policy then, typically, $y_s$ and $y_t$ are good actions for $s$ and $t$ respectively. Hence, by the inductive bias and since $s$ is close to $t$ we often have that $\mathbb{E}[\ell_{t,y_s}]$ is similar to $\mathbb{E}[\ell_{t,y_t}]$. Hence, $\mu_t$ is often close to 0. However, since the setting is fully adversarial, it may not be.

We note that, unlike NN, our algorithm HNN is also adaptive to factors that influence the values of the excesses, such as the local Holder complexity and its constant in i.i.d. stochastic bandits (as defined in Section 4.5).

## 6 Experiments

In Figure 3 we give the results of two real-world experiments comparing the performance of HNN to that of NN and three state of the art algorithms for the i.i.d. stochastic problem (Figure 3a) and a real-world adversarial problem (Figure 3b). It is important to note that the first of these experiments is on an i.i.d. stochastic dataset, whereas HNN was built for the more general adversarial problem, yet in both experiments HNN manages to outperform the algorithms designed specifically for the i.i.d. stochastic problem.

The details of the experiments, which are both for problems in cyber-defence, are given in Appendix B. We note that in the first experiment, NN and NN with a binning radius of 0.1 achieve the best performance, HNN attains the next-best results, surpassing the other three bandits and NN with a binning radius of 0.2. In the second experiment, HNN achieves the best performance, even when compared with NN with various binning radii.

## 7 Conclusion

Inspired by Pasteris et al. (2023) and Slivkins (2009), we designed a novel algorithm for the adversarial bandit problem in metric spaces. Unlike the algorithm of Pasteris et al. (2023), our algorithm is locally adaptive to

the density of contexts and the smoothness of the decision boundary of the comparator policy. We analysed the regret of our algorithm with respect to any possible comparator policy, and analysed further the special case of Euclidean space, for both adversarial and stochastic bandits. We gave the results of real-world experiments that empirically demonstrate the performance of our algorithm.

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

## A  Notation Table

We now give a table of some of the notation used in this paper.

| Symbol | Meaning |
| --- | --- |
| $\mathbb{R}$ | Set of all real numbers |
| $\mathbb{N}$ | Set of all natural numbers excluding 0 |
| $[n]$ | $\{m \in \mathbb{N} \mid m \leq n\}$ |
| $\llbracket P \rrbracket$ | Equal to 1 if predicate $P$ is true and equal to 0 otherwise |
| $x_t$ | Context at trial $t$ (usually considered implicit) |
| $\ell_{t,a}$ | Loss of action $a$ on trial $t$ |
| $a_t$ | Action selected by the algorithm on trial $t$ |
| $\Delta_{s,t}$ | Distance from $x_s$ to $x_t$ |
| $\epsilon$ | Binning radius |
| $c$ | Approximation quality of nearest neighbour search algorithm |
| $\rho$ | Learning rate (parameter of the algorithm) |
| $T$ | Number of trials |
| $K$ | Number of actions |
| $d$ | Dimensionality of Euclidean space |
| $\boldsymbol{y}$ | An arbitrary policy in $[K]^T$, which associates an action $y_t$ with each trial $t$ |
| $R(\boldsymbol{y})$ | Regret of the algorithm with respect to policy $\boldsymbol{y}$ |
| $\mathcal{T}$ | Growing tree (with trials as vertices) constructed by the algorithm |
| $\mathcal{X}$ | Euclidean ball centered at $\boldsymbol{0}$ with radius $1/2$ |

## B  Experiment Details

We ran experiments on two real-life firewall datasets to empirically compare HNN with NN and with three other state of the art contextual bandit algorithms, *K-Nearest Neighbours with UCB* (KNN_UCB), *K-Nearest Neighbours with KL divergence and UCB* (KNN_KL_UCB) Reeve et al. (2018) and *contextual bandits with similarity information* (Slivkin's) Slivkins (2009). It is important to note that the first experiment is on an i.i.d. stochastic dataset, whereas HNN was built for the adversarial case, yet in both experiments HNN manages to outperform algorithms designed specifically for the i.i.d. stochastic problem.

Both sets of experiments involved using classification datasets. The loss was determined by whether the action chosen matched the label from the dataset, with a loss of 1 if it did not, and 0 otherwise.

For our first experiment, we used the UCI ML Firewall dataset int (2019), which is internet firewall data. We randomly shuffled and ran the dataset multiple times to get the average cumulative loss of each algorithm. We used the Action column as the actions for the bandits. The dataset includes the actions `Allow`, `Block`, `Drop` and `Reset-All`, but we removed the `Reset-All` data due to the extreme sparsity of this class (less than 1% of the dataset). We used the following features: `SourcePort`, `DestinationPort`, `NATSourcePort`, `NATDestinationPort`, `BytesSent`, `pkts_sent`. We removed features that we believed would not be accessible at the point of decision for a bandit if it was being utilised in the real-world for intrusion detection (e.g. features such as `ElapsedTime`).

For the distance metric, we took the Euclidean difference of each of the non-categorical features. For the categorical features (the Port features) we used a distance measure of 0 if the value was identical and 1 if it was not. For each feature, we independently scaled the pairwise differences to the $[0, 1/6]$ range (6 being the total number of features), to ensure that all features contributed equally to the overall distance computation (i.e. we had no weightings across the features). The final distances were constrained to the $[0, 1]$ range.

We compared HNN with the other contextual bandits with no binning. We also compared against NN with binning radii of 0.1 and 0.2. The results are shown in Figure 3a, noting that the standard deviation for all but *KNN_KL_UCB* are very minimal and as such not visible on the plot. NN with a binning radius of 0.1 has the lowest cumulative loss, followed by NN and then closely by HNN. Both NN and HNN outperform the other state of the art bandits, even in this stochastic setting.

For the second experiment, we used another real-world dataset, using a subset of the CICIDS2017 intrusion dataset Sharafaldin et al. (2018). For this experiment, we used the machine learning CVE data, which consisted of the network traffic flows. The CICIDS2017 dataset has multiple days of data, on each day, there is benign and attack traffic. We took a subset of the Wednesday data which contained multiple different DoS attacks, we maintained the temporal order from the original dataset so that the experiment was conducted in the adversarial setting.

We gave the bandits two actions, `Allow` or `Block`, and we used the label from the dataset to determine which action was most appropriate (allow for benign, block for attack). We used all 77 features of the dataset (dropping the duplicate column of "Fwd Header Length").

We used a similar distance measure as for the first dataset, where the categorical data (IP address) had a distance of 0 if it was the same or 1 otherwise, and for the non-categorical features, we took the Euclidean difference. We again independently scaled the pairwise differences to the $[0, 1/77]$ range, so that all features contributed equally to the overall distance computation. The overall distances were again constrained to the $[0, 1]$ range.

Figure 3b shows that HNN clearly outperforms all the other algorithms, including NN both without binning and also with binning radii of 0.1 and 0.05.

## C Proof of Theorem 5.2

We now prove Theorem 5.2. We note that we use the results of Pasteris et al. (2023) as a black box in this analysis.

Without loss of generality we assume that for all $s, t \in [T]$ with $s \neq t$ we have $\Delta_{s,t} \neq 0$ which, since $\epsilon = 0$, implies that the binning step in the construction of $\mathcal{T}$ is never invoked. This is without loss of generality as if there was to exist $s > t$ with $\Delta_{s,t} = 0$ we simply define $\hat{y}_s := \hat{y}_t$ in this proof. Also without loss of generality assume there exists $s, t \in [T] \setminus \mathcal{M}$ with $y_s \neq y_t$ else the result is trivial (by choosing $\hat{y}_t$ to be the same for all $t \in [T]$)

Let $\phi$ be an arbitrary number in $(0, 1)$. We define:

$$f := 1/2 \quad ; \quad \beta := 2/\phi$$

$$\lambda := (\phi\beta + (1+\beta)/f)/(\beta(1-\phi))$$

and:

$$z := (1-f)f/(2c(1+\beta)).$$

In this analysis we will often use the fact that, for all $t \in [T]$, we have $d_{p_t} = d_t - 1$ and $\Delta_{t,p_t} \leq f^{d_t - 1}$.

**Definition C.1.** Consider the rooted tree with vertex set $[T]$ such that, for all $t \in [T] \setminus \{1\}$, we have that $p_t$ is the parent of $t$. Let $\mathcal{L}$ be the set of leaves of this tree. Given $t \in [T]$ we then define $\mathcal{D}_t$ to be the set of all descendants of $t$ and define $\mathcal{A}_t$ to be the set of all ancestors of $t$.

**Lemma C.2.** *For all $r, t \in [T]$ with $r \neq t$ and $d_r = d_t$ we have that $\Delta_{r,t} > f^{d_r}/c$.*

*Proof.* Suppose, for contradiction, the converse: that $\Delta_{r,t} \leq f^{d_r}/c$. Without loss of generality assume $r < t$. Let $h := \max\{d_s \mid s \in [t-1]\}$ and for all $d \in [h]$ let $s_d$ be as created by the algorithm on trial $t$. Let $q := s_{d_r}$. Since $q$ is a $c$-nearest neighbour of $t$ in the set $\{s \in [t-1] \mid d_s = d_r\}$ (which contains $r$) we must have that $\Delta_{q,t} \leq c\Delta_{r,t} \leq f^{d_r}$. But from the algorithm we have that $d_t - 1$ is the maximum value of $d \in [h]$ such that $\Delta_{s_d,t} \leq f^d$ so since $d_t - 1 = d_r - 1 < d_r$ we have a contradiction. $\qquad\square$

**Definition C.3.** Define $\mathcal{U}$ to be the set of all trials $t \in [T]$ in which for all $r, s \in [T] \setminus \mathcal{M}$ with $\Delta_{r,t} \leq \beta f^{d_t}$ and $\Delta_{s,t} \leq \beta f^{d_t}$ we have $y_r = y_s$.

**Lemma C.4.** *Given $s, t \in [T]$ with $s \in \mathcal{U}$ and $t \in \mathcal{D}_s$ we have $t \in \mathcal{U}$.*

*Proof.* Noting that $d_t \geq d_s$ we fix $s$ and prove by induction on $d_t$. When $d_t = d_s$ we have $t = s$ so the result is immediate. Now suppose, for some $d \geq d_s$, that the inductive hypothesis holds for all $t$ with $d_t = d$. Now take $t$ with $d_t = d + 1$. Since $t \in \mathcal{D}_s$ and $t \neq s$ we have $p_t \in \mathcal{D}_s$. So since $d_{p_t} = d$ we have, by the inductive hypothesis, that $p_t \in \mathcal{U}$. Now take any $q, r \in [T] \setminus \mathcal{M}$ with $\Delta_{q,t} \leq \beta f^{d_t}$ and $\Delta_{r,t} \leq \beta f^{d_t}$. From the algorithm we have that $\Delta_{t,p_t} \leq f^{d_t - 1}$ and hence, by the triangle inequality, we have:

$$\Delta_{r,p_t} \leq \Delta_{r,t} + \Delta_{t,p_t} \leq \beta f^{d_t} + f^{d_t - 1} = (\beta f + 1)f^{d_t - 1} \leq \beta f^{d_t - 1}$$

Similarly we have $\Delta_{q,p_t} \leq \beta f^{d_t - 1}$. So since $d_{p_t} = d_t - 1$ and $p_t \in \mathcal{U}$ we must have that $y_q = y_r$. Hence, we must have that $t \in \mathcal{U}$ so the result holds by induction. $\qquad\square$

**Definition C.5.** Let $\mathcal{V}$ be the set of all $t \in [T]$ such that either:

- $t \in \mathcal{U}$ and $p_t \notin \mathcal{U}$

- $t \in \mathcal{L}$ and $t \notin \mathcal{U}$

**Definition C.6.** Let $\mathcal{W}$ be the set of all $t \in \mathcal{V}$ such that there does not exist $s \in \mathcal{V}$ with $d_s > d_t$ and $\Delta_{s,t} \leq (f^{d_t} - f^{d_s})/(2c)$

**Definition C.7.** For any $t \in \mathcal{W}$ let $\mathcal{Q}_t$ be equal to the set of all $s \in \mathcal{V} \setminus \mathcal{W}$ in which $\Delta_{s,t} \leq f^{d_s}/(2c)$

**Lemma C.8.** *Given $s \in \mathcal{V} \setminus \mathcal{W}$ and $d \in \mathbb{N}$ we either have that there exists $t \in \mathcal{W}$ with $s \in \mathcal{Q}_t$ or that there exists some $r \in \mathcal{V} \setminus \mathcal{W}$ with $d_r \geq d$ and $\Delta_{r,s} \leq (f^{d_s} - f^{d_r})/(2c)$*

*Proof.* If there exists $t \in \mathcal{W}$ with $s \in \mathcal{Q}_t$ then we're done so assume otherwise. We prove by induction on $d$. We immediately have the result for $d = 0$ by choosing $r := s$. Now suppose, for some $d' \in \mathbb{N} \cup \{0\}$, that the inductive hypothesis holds when $d = d'$ and consider the case that $d = d' + 1$. By the inductive hypothesis choose $q \in \mathcal{V} \setminus \mathcal{W}$ with $d_q \geq d'$ and $\Delta_{q,s} \leq (f^{d_s} - f^{d_q})/(2c)$. Since $q \in \mathcal{V} \setminus \mathcal{W}$ we have, by definition of $\mathcal{W}$, that there exists $u \in \mathcal{V}$ with $d_u > d_q$ and $\Delta_{u,q} \leq (f^{d_q} - f^{d_u})/(2c)$. By the triangle inequality we then have:

$$\Delta_{u,s} \leq \Delta_{u,q} + \Delta_{q,s} \leq (f^{d_q} - f^{d_u})/(2c) + (f^{d_s} - f^{d_q})/(2c) = (f^{d_s} - f^{d_u})/(2c)$$

If it was the case that $u \in \mathcal{W}$ we would have, from this inequality, that $s \in \mathcal{Q}_u$ which is a contradiction. Hence, we have $u \in \mathcal{V} \setminus \mathcal{W}$. Since $d_u > d_q$ and $d_q \geq d'$ we have $d_u \geq d' + 1 = d$. By the above inequality we then have the result by choosing $r := u$. This completes the inductive proof. $\qquad\square$

**Lemma C.9.** *Given $s \in \mathcal{V} \setminus \mathcal{W}$ there exists $t \in \mathcal{W}$ with $s \in \mathcal{Q}_t$.*

*Proof.* Suppose, for contradiction, the converse. By Lemma C.8 we then have, for all $d \in \mathbb{N}$, that there exists some $r \in \mathcal{V} \setminus \mathcal{W}$ with $d_r \geq d$. By choosing $d := T$ we then have that there exists $r \in [T]$ with $d_r \geq T$ which is impossible. $\square$

**Lemma C.10.** *For all $t \in \mathcal{W}$ and $s \in \mathcal{Q}_t$ we have $d_s \leq d_t$.*

*Proof.* Suppose, for contradiction, that there exists $t \in \mathcal{W}$ and $s \in \mathcal{Q}_t$ with $d_s > d_t$. Then by definition of $\mathcal{Q}_t$ we have $s \in \mathcal{V}$ and $\Delta_{s,t} \leq f^{d_s}/(2c)$. Since $d_t \leq d_s - 1$ we then have:

$$(f^{d_t} - f^{d_s})/(2c) \geq (f^{d_s-1} - f^{d_s})/(2c) = (1/f - 1)f^{d_s}/(2c) \geq (1/f - 1)\Delta_{s,t}$$

So since $f = 1/2$ we have $\Delta_{s,t} \leq (f^{d_t} - f^{d_s})/(2c)$ which, since $d_s > d_t$ and $s \in \mathcal{V}$, contradicts the fact that $t \in \mathcal{W}$. $\square$

**Lemma C.11.** *For all $t \in \mathcal{W}$ we have $|\mathcal{Q}_t| \leq d_t + 1$.*

*Proof.* By Lemma C.10 all we need to prove is that if $q, r \in \mathcal{Q}_t$ are such that $q \neq r$ then $d_q \neq d_r$. We now prove this by considering the converse: that $d_q = d_r$. By definition of $\mathcal{Q}_t$ we have $\Delta_{q,t} \leq f^{d_q}/(2c)$ and $\Delta_{r,t} \leq f^{d_r}/(2c)$ so by the triangle inequality we have:

$$\Delta_{q,r} \leq \Delta_{q,t} + \Delta_{r,t} \leq f^{d_q}/(2c) + f^{d_r}/(2c) = f^{d_q}/c$$

which, since $d_q = d_r$, contradicts Lemma C.2. $\square$

**Lemma C.12.** *For all $t \in \mathcal{V}$ we have $f^{d_t} \geq \gamma_t f/(1 + \beta)$*

*Proof.* By definition of $\mathcal{V}$ we immediately have that either $t \notin \mathcal{U}$ or $p_t \notin \mathcal{U}$. By Lemma C.4 we then have that $p_t \notin \mathcal{U}$. Hence, by definition of $\mathcal{U}$ and since $d_{p_t} = d_t - 1$, we can choose $r, s \in [T] \setminus \mathcal{M}$ with $y_r \neq y_s$ and $\Delta_{r,p_t} \leq \beta f^{d_t-1}$ and $\Delta_{s,p_t} \leq \beta f^{d_t-1}$. Since $y_r \neq y_s$ we can, without loss of generality, assume that $y_s \neq y_t$ which means, since $s \notin \mathcal{M}$, that $\Delta_{s,t} \geq \gamma_t$. By the triangle inequality and the fact that $\Delta_{t,p_t} \leq f^{d_t-1}$ we then have:

$$\gamma_t \leq \Delta_{s,t} \leq \Delta_{s,p_t} + \Delta_{p_t,t} \leq \beta f^{d_t-1} + f^{d_t-1} = (1 + \beta)f^{d_t}/f$$

Rearranging then gives us the desired result. $\square$

**Lemma C.13.** *For all $s, t \in \mathcal{W}$ with $s \neq t$ we have $\Delta_{s,t} > z \max(\gamma_s, \gamma_t)$.*

*Proof.* Without loss of generality assume $d_s \geq d_t$. If $d_s = d_t$ then we have, from Lemma C.2, that $\Delta_{s,t} > f^{d_t}/c$. On the other hand, if $d_s > d_t$ then we have, from the definition of $\mathcal{W}$ and the fact that $s \in \mathcal{V}$, that:

$$\Delta_{s,t} > (f^{d_t} - f^{d_s})/(2c) \geq (f^{d_t} - f^{d_t+1})/(2c) = (1 - f)f^{d_t}/(2c)$$

In either case we have that $\Delta_{s,t} > (1 - f)f^{d_t}/(2c)$ and hence, since $f^{d_s} \leq f^{d_t}$, we also have that $\Delta_{s,t} > (1 - f)f^{d_s}/(2c)$. From Lemma C.12 we then have that:

$$\Delta_{s,t} > (1 - f)f^{d_t}/(2c) \geq \gamma_t(1 - f)f/(2c(1 + \beta)) = z\gamma_t$$

and

$$\Delta_{s,t} > (1 - f)f^{d_s}/(2c) \geq \gamma_s(1 - f)f/(2c(1 + \beta)) = z\gamma_s$$

as required. $\square$

**Lemma C.14.** *We have $|\mathcal{W}| \leq \Psi$*

*Proof.* Immediate from Lemma C.13 and the definition of $\Psi$. $\square$

**Lemma C.15.** *For all $t \in \mathcal{V}$ we have $d_t \in \mathcal{O}(\ln(1/\Lambda))$*

*Proof.* By Lemma C.12 and definition of $\Lambda$ we have:

$$f^{d_t} \geq \gamma_t f/(1+\beta) \geq \Lambda f/(1+\beta)$$

Taking logarithms gives us the result. $\qquad\square$

**Lemma C.16.** *We have $|\mathcal{V}| \in \mathcal{O}(\Psi \ln(1/\Lambda))$*

*Proof.* By Lemma C.9 we have:

$$\mathcal{V} = \mathcal{W} \cup \bigcup_{t \in \mathcal{W}} \mathcal{Q}_t$$

so that:

$$|\mathcal{V}| \leq |\mathcal{W}| + \sum_{t \in \mathcal{W}} |\mathcal{Q}_t|$$

By lemmas C.11 and C.15 we have $|\mathcal{Q}_t| \in \mathcal{O}(\ln(1/\Lambda))$ for all $t \in \mathcal{W}$. Substituting into the above inequality gives us $|\mathcal{V}| \leq \mathcal{O}(|\mathcal{W}| \ln(1/\Lambda))$. Lemma C.14 then gives us the result. $\qquad\square$

**Lemma C.17.** *Suppose we have some $t \in [T]$ such that for all $s \in \mathcal{A}_t$ we have $s \notin \mathcal{V}$. Then $t \notin \mathcal{U}$.*

*Proof.* We prove by induction on $d_t$. If $d_t = 0$ then we have $t = 1$ so that we immediately have $t \notin \mathcal{U}$ (since there exists $r, s \in [T] \setminus \mathcal{M}$ with $y_r \neq y_s$). Given some $d > 0$ suppose that the inductive hypothesis holds for all $t$ with $d_t = d$. Now consider any $t$ with $d_t = d + 1$. Note that for all $s \in \mathcal{A}_{p_t}$ we have $s \in \mathcal{A}_t$ so that $s \notin \mathcal{V}$. Since $d_{p_t} = d_t - 1 = d$ we then have, by the inductive hypothesis, that $p_t \notin \mathcal{U}$. If it was the case that $t \in \mathcal{U}$ we would then have, by definition of $\mathcal{V}$, that $t \in \mathcal{V}$. But since $t \in \mathcal{A}_t$ this would be a contradiction. Hence, $t \notin \mathcal{U}$. This completes the inductive proof. $\qquad\square$

**Lemma C.18.** *For all $t \in [T]$ there exists $s \in \mathcal{V}$ such that $t \in \mathcal{D}_s \cup \mathcal{A}_s$.*

*Proof.* Assume, for contradiction, the converse: that there exists no $s \in \mathcal{V}$ with $t \in \mathcal{D}_s \cup \mathcal{A}_s$. This means that for all $s \in \mathcal{D}_t \cup \mathcal{A}_t$ we have $s \notin \mathcal{V}$. So choose some $r \in \mathcal{D}_t \cap \mathcal{L}$. Since $\mathcal{A}_r \subseteq \mathcal{D}_t \cup \mathcal{A}_t$ we have, for all $s \in \mathcal{A}_r$, that $s \notin \mathcal{V}$. By Lemma C.17 we hence have that $r \notin \mathcal{U}$. But since $r \in \mathcal{L}$ this would mean that $r \in \mathcal{V}$ which, since $t \in \mathcal{D}_r \cup \mathcal{A}_r$, is a contradiction. $\qquad\square$

**Definition C.19.** Define the policy $\hat{\boldsymbol{y}} \in [K]^T$ such that:

- If $t \notin \mathcal{U}$ then $\hat{y}_t := y_t$.

- If $t \in \mathcal{U}$ and there exists $s \in [T] \setminus \mathcal{M}$ with $\Delta_{s,t} \leq \beta f^{d_t}$ then $\hat{y}_t = y_s$. Note that by definition of $\mathcal{U}$ we have that $\hat{y}_t$ is uniquely defined.

- If $t \in \mathcal{U}$ and there does not exist $s \in [T] \setminus \mathcal{M}$ with $\Delta_{s,t} \leq \beta f^{d_t}$, we have $\hat{y}_t := \hat{y}_{p_t}$. Since $1 \notin \mathcal{U}$ this is defined.

**Lemma C.20.** *Given $t \in [T] \setminus \{1\}$ with $\hat{y}_t \neq \hat{y}_{p_t}$ there exists $s \in \mathcal{V}$ with $t \in \mathcal{A}_s$.*

*Proof.* By Lemma C.18 choose $s \in \mathcal{V}$ such that $t \in \mathcal{D}_s \cup \mathcal{A}_s$. Assume, for contradiction, that $t \notin \mathcal{A}_s$. Then we must have $t \in \mathcal{D}_s \setminus \{s\}$. This means that $s \notin \mathcal{L}$ and hence, by definition of $\mathcal{V}$, we have that $s \in \mathcal{U}$. So since we have both $t \in \mathcal{D}_s$ and $p_t \in \mathcal{D}_s$ we have, by Lemma C.4, that both $t \in \mathcal{U}$ and $p_t \in \mathcal{U}$. Since $\hat{y}_t \neq \hat{y}_{p_t}$ we must then have, by definition of $\hat{\boldsymbol{y}}$, that there exists $q \in [T] \setminus \mathcal{M}$ with $\Delta_{q,t} \leq \beta f^{d_t}$. Since $\Delta_{t,p_t} \leq f^{d_t - 1}$ we have, by the triangle inequality, that:

$$\Delta_{q,p_t} \leq \Delta_{q,t} + \Delta_{t,p_t} \leq \beta f^{d_t} + f^{d_t - 1} = (\beta f + 1) f^{d_t - 1} \leq \beta f^{d_t - 1}$$

so since $d_{p_t} = d_t - 1$ and $q \notin \mathcal{M}$ we have, by definition of $\hat{\boldsymbol{y}}$ and since $p_t \in \mathcal{U}$, that $\hat{y}_{p_t} = y_q$. We also have, by definition of $\hat{\boldsymbol{y}}$ and since both $\Delta_{q,t} \leq \beta f^{d_t}$ and $t \in \mathcal{U}$, that $\hat{y}_t = y_q$. But this means that $\hat{y}_t = \hat{y}_{p_t}$ which is a contradiction. We have hence shown that $t \in \mathcal{A}_s$. $\qquad\square$

**Lemma C.21.** *We have:*
$$\sum_{t \in [T] \setminus \{1\}} [\![\hat{y}_t \neq \hat{y}_{p_t}]\!] \in \mathcal{O}(\Psi \ln(1/\Lambda)^2)$$

*Proof.* Given $t \in \mathcal{V}$ we have, by Lemma C.15, that $d_t \in \mathcal{O}(\ln(1/\Lambda))$ and hence that $|\mathcal{A}_t| \in \mathcal{O}(\ln(1/\Lambda))$. By lemmas C.20 and C.16 we then have that:

$$\sum_{t \in [T] \setminus \{1\}} [\![\hat{y}_t \neq \hat{y}_{p_t}]\!] \leq \left| \bigcup_{s \in \mathcal{V}} \mathcal{A}_s \right| \leq \sum_{s \in \mathcal{V}} |\mathcal{A}_s| \in \mathcal{O}(|\mathcal{V}| \ln(1/\Lambda)) \in \mathcal{O}(\Psi \ln(1/\Lambda)^2)$$

as required. $\square$

**Lemma C.22.** *We have:*

$$\sum_{t \in [T]} \mathbb{E}[\ell_{t,a_t} - \ell_{t,\hat{y}_t}] \leq \tilde{\mathcal{O}} \left( \left( \rho - \frac{\Psi \ln(1/\Lambda)^2}{\rho} \right) \sqrt{KT} \right)$$

*Proof.* From Pasteris et al. (2023), when using either belief propagation or CBNN, we have, when taking expectations on their result, that:

$$\sum_{t \in [T]} \mathbb{E}[\ell_{t,a_t} - \ell_{t,\hat{y}_t}] \in \tilde{\mathcal{O}} \left( \left( \rho - \frac{1 + \sum_{t \in [T] \setminus \{1\}} [\![\hat{y}_t \neq \hat{y}_{p_t}]\!]}{\rho} \right) \sqrt{KT} \right)$$

Substituting in Lemma C.21 gives us the result. $\square$

**Lemma C.23.** *For all $t \in [T]$ with $\hat{y}_t \neq y_t$ we have $t \in \mathcal{M} \cap \mathcal{U}$.*

*Proof.* Suppose, for contradiction, that there does exist $t \in [T]$ with $t \notin \mathcal{M} \cap \mathcal{U}$ and $\hat{y}_t \neq y_t$. Since $\hat{y}_t \neq y_t$ we must have, by definition of $\hat{\boldsymbol{y}}$, that $t \in \mathcal{U}$. Hence, we must also have $t \notin \mathcal{M}$. Since both $t \in \mathcal{U}$ and $t \notin \mathcal{M}$ with $\Delta_{t,t} = 0 \leq \beta f^{d_t}$ we must have, by definition of $\hat{\boldsymbol{y}}$, that $\hat{y}_t = y_t$ which is a contradiction. $\square$

**Lemma C.24.** *For all $s, t \in [T]$ with $s \in \mathcal{D}_t$ we have $\Delta_{s,t} \leq \phi \beta f^{d_t}$*

*Proof.* We hold $s$ fixed and prove by reverse induction on $d_t$ (i.e. from $d_s$ to 0). When $d_t = d_s$ we have $s = t$ and hence $\Delta_{s,t} = 0$ so the result holds trivially. Now suppose, for some $d \in [d_s]$, that the inductive hypothesis holds when $d_t = d$. We now show that it holds when $d_t = d - 1$ which will complete the inductive proof. So take $t$ with $d_t = d - 1$. Let $r$ be such that $s \in \mathcal{D}_r$ and $p_r = t$. Note that we have $d_r = d$ so by the inductive hypothesis we have $\Delta_{s,r} \leq \phi \beta f^d$. Since $\Delta_{r,p_r} \leq f^{d_r - 1}$ we then have, by the triangle inequality, that:
$$\Delta_{s,t} \leq \Delta_{s,r} + \Delta_{r,t} = \Delta_{s,r} + \Delta_{r,p_r} \leq \phi \beta f^d + f^{d-1} = (\phi \beta f + 1) f^{d-1} = \phi \beta f^{d-1}$$
since $\beta = 2/\phi$ and $f = 1/2$. $\square$

**Lemma C.25.** *For all $t \in \mathcal{M} \cap \mathcal{U}$ we have $\hat{y}_t \in \mathcal{Y}_t$.*

*Proof.* Let $\mathcal{S}$ be the set of all $r \in [T]$ such that there exists $q \in [T] \setminus \mathcal{M}$ with $\Delta_{q,r} \leq \beta f^{d_r}$. Define:

$$s := \operatorname{argmin}_{r \in \mathcal{A}_t \cap \mathcal{U}} d_r \quad ; \quad v := \operatorname{argmax}_{r \in \mathcal{A}_t \cap \mathcal{S}} d_r$$

noting that these both exist since $t \in \mathcal{A}_t \cap \mathcal{U}$ and $1 \in \mathcal{A}_t \cap \mathcal{S}$. Since $1 \notin \mathcal{U}$ (which comes directly from the fact that there exists $q, r \in [T] \setminus \mathcal{M}$ with $y_q \neq y_r$) we have that $s \neq 1$ and hence $p_s$ exists so let $d := d_{p_s}$. Since $p_s \in \mathcal{A}_t$ with $d_{p_s} < d_s$, we have $p_s \notin \mathcal{U}$ so by definition of $\mathcal{U}$ there exists $r \in [T] \setminus \mathcal{M}$ with $\Delta_{r,p_s} \leq \beta f^d$. This means that $p_s \in \mathcal{S}$ and hence that $v \in \mathcal{D}_{p_s}$. Define:

$$w := \operatorname{argmin}_{q \in [T] \setminus \mathcal{M}} \Delta_{q,t}$$

so that $\Delta_{w,t} = \theta_t$.

We have two cases. First consider the case that $v = t$. In this case we have $t \in \mathcal{S}$ so that there exists $q \in [T] \setminus \mathcal{M}$ such that $\Delta_{q,t} \le \beta f^{d_t}$. By definition of $w$ we have that $\Delta_{w,t} \le \Delta_{q,t}$ so $\Delta_{w,t} \le \beta f^{d_t}$ and hence, by definition of $\hat{\boldsymbol{y}}$ and since $t \in \mathcal{U}$ and $w \notin \mathcal{M}$, we have $\hat{y}_t = y_w$. Since $\Delta_{w,t} < \lambda \theta_t$ we have, by definition of $\mathcal{Y}_t$, that $y_w \in \mathcal{Y}_t$. Hence, $\hat{y}_t \in \mathcal{Y}_t$ as required.

Next consider the case that $v \neq t$. Choose $u \in [T]$ as follows:

- If $v = p_s$ then since $p_s \notin \mathcal{U}$, choosing $u = v$ gives us, by definition of $\hat{\boldsymbol{y}}$, that $\hat{y}_v = y_u$.

- If $v \neq p_s$ then, since $v \in \mathcal{D}_{p_s}$ and $s, v \in \mathcal{A}_t$, we have $v \in \mathcal{D}_s$ so, since $s \in \mathcal{U}$, we have, by Lemma C.4 that $v \in \mathcal{U}$. So $v \in \mathcal{U} \cap \mathcal{S}$ and so, by definition of $\hat{\boldsymbol{y}}$ and $\mathcal{S}$, choose $u \in [T]$ such that $\Delta_{u,v} \le \beta f^{d_v}$ and $\hat{y}_v = y_u$.

Either way, we have $\Delta_{u,v} \le \beta f^{d_v}$ and $\hat{y}_v = y_u$. Since $v \in \mathcal{A}_t \setminus \{t\}$ let $q \in \mathcal{A}_t$ be such that $p_q = v$. We have, by Lemma C.24 and the triangle inequality, that:

$$\Delta_{w,q} \le \Delta_{w,t} + \Delta_{t,q} \le \theta_t + \phi \beta f^{d_q}$$

Since $q \in \mathcal{A}_t$ and $d_q > d_v$ we must have, by definition of $v$, that $q \notin \mathcal{S}$ so since $w \notin \mathcal{M}$ we also have:

$$\Delta_{w,q} > \beta f^{d_q}$$

Substituting this inequality into the previous and rearranging gives us:

$$\theta_t > \beta f^{d_q} - \phi \beta f^{d_q} = \beta(1 - \phi) f^{d_q}$$

so that:

$$f^{d_q} < \theta_t / (\beta(1 - \phi))$$

Since $d_v = d_q - 1$ and $\Delta_{q,v} \le f^{d_v}$ (as $v = p_q$) we then have, from the triangle inequality and Lemma C.24, that:

$$\begin{aligned} \Delta_{t,u} \le \Delta_{t,q} + \Delta_{q,v} + \Delta_{v,u} &\le \phi \beta f^{d_q} + f^{d_v} + \beta f^{d_v} \\ &= (\phi \beta + (1 + \beta)/f) f^{d_q} \\ &< (\phi \beta + (1 + \beta)/f) \theta_t / (\beta(1 - \phi)) \\ &= \lambda \theta_t \end{aligned}$$

So that $y_u \in \mathcal{Y}_t$. Since $\hat{y}_v = y_u$, all that is left to do now is to prove that $\hat{y}_t = \hat{y}_v$. To prove this we need only show that for all $r \in (\mathcal{D}_v \setminus \{v\}) \cap \mathcal{A}_t$ we have $\hat{y}_r = \hat{y}_{p_r}$. To show this take any such $r \in (\mathcal{D}_v \setminus \{v\}) \cap \mathcal{A}_t$. Since $v \in \mathcal{D}_{p_s}$ and $s, v \in \mathcal{A}_t$ we must have $r \in \mathcal{D}_s$ and hence, by Lemma C.4 and the fact that $s \in \mathcal{U}$, we have $r \in \mathcal{U}$. Since $d_r > d_v$ and $r \in \mathcal{A}_t$ we have, by definition of $v$, that $r \notin \mathcal{S}$. So $r \in \mathcal{U} \setminus \mathcal{S}$ and hence, by definition of $\hat{\boldsymbol{y}}$ and $\mathcal{S}$, we have $\hat{y}_r = \hat{y}_{p_r}$ as required. $\square$

**Lemma C.26.** *We have:*

$$\sum_{t \in [T]} \mathbb{E}[\ell_{t,\hat{y}_t} - \ell_{t,y_t}] \le \sum_{t \in \mathcal{M}} \mu_t$$

*Proof.* By Lemma C.23 we have:

$$\sum_{t \in [T] \setminus \mathcal{M}} \mathbb{E}[\ell_{t,\hat{y}_t} - \ell_{t,y_t}] = \sum_{t \in [T] \setminus \mathcal{M}} \mathbb{E}[\ell_{t,y_t} - \ell_{t,y_t}] = 0$$

By Lemma C.23 we have, for all $t \in \mathcal{M} \setminus \mathcal{U}$, that $\hat{y}_t = y_t \in \mathcal{Y}_t$. So by Lemma C.25 we have, for all $t \in \mathcal{M}$, that $\hat{y}_t \in \mathcal{Y}_t$. Hence, we have:

$$\sum_{t \in \mathcal{M}} \mathbb{E}[\ell_{t,\hat{y}_t} - \ell_{t,y_t}] \le \sum_{t \in \mathcal{M}} \max_{a \in \mathcal{Y}_t} \mathbb{E}[\ell_{t,a} - \ell_{t,y_t}] = \sum_{t \in \mathcal{M}} \mu_t$$

Combining these two (in)equalities gives us:

$$\sum_{t\in[T]}\mathbb{E}[\ell_{t,\hat{y}_t}-\ell_{t,y_t}]=\sum_{t\in[T]\setminus\mathcal{M}}\mathbb{E}[\ell_{t,\hat{y}_t}-\ell_{t,y_t}]+\sum_{t\in\mathcal{M}}\mathbb{E}[\ell_{t,\hat{y}_t}-\ell_{t,y_t}]\leq 0+\sum_{t\in\mathcal{M}}\mu_t=\sum_{t\in\mathcal{M}}\mu_t$$

as required. $\qquad\square$

**Lemma C.27.** *We have:*

$$R(\boldsymbol{y})\leq\sum_{t\in\mathcal{M}}\mu_t+\tilde{\mathcal{O}}\left(\left(\rho+\frac{\Psi\ln(1/\Lambda)^2}{\rho}\right)\sqrt{KT}\right).$$

*Proof.* By linearity of expectations we have:

$$R(\boldsymbol{y})=\sum_{t\in[T]}\mathbb{E}[\ell_{t,a_t}-\ell_{t,y_t}]=\sum_{t\in[T]}\mathbb{E}[\ell_{t,a_t}-\ell_{t,\hat{y}_t}]+\sum_{t\in[T]}\mathbb{E}[\ell_{t,\hat{y}_t}-\ell_{t,y_t}]$$

Substituting in lemmas C.22 and C.26 then gives us the result. $\qquad\square$

This completes the proof. $\qquad\blacksquare$

## D  Proof of Theorem 4.1

We first note that the only effect that $\epsilon$ has on the bound is that the contexts can be moved by a distance of up to $\epsilon$. Since $\epsilon\leq(1-\xi)\min_{i\in[N]}r_i/2$ and $\xi>1$ is arbitrary when can hence assume, without loss of generality, that $\epsilon=0$. We can hence apply Theorem 5.2 with our choice of margin $\mathcal{M}$.

Let $\tilde{y}$ be an extension of $\boldsymbol{y}$ such that $\{\mathcal{B}(v_i,r_i)\,|\,i\in[N]\}$ covers the decision boundary of $\tilde{y}$. Let $\mathcal{D}$ be the decision boundary of $\tilde{y}$.

**Definition D.1.** For all $t\in[T]$ let $q_t$ be the minimiser of $\Delta_{s,t}$ out of all $s\in[T]\setminus\mathcal{M}$ with $y_s\neq y_t$. Since $\tilde{y}(x_t)\neq\tilde{y}(x_{q_t})$ choose $b_t\in\mathcal{D}$ such that $b_t$ lies on the straight line from $x_t$ to $x_{q_t}$. Since $b_t\in\mathcal{D}$ choose $i_t\in[N]$ such that $b_t\in\mathcal{B}(v_{i_t},r_{i_t})$.

**Definition D.2.** Define $J:=C\log_2(T)$. For all $t\in[T]$ define $j_t$ as the minimum number in $[J]\cup\{0\}$ such that $x_t\in\mathcal{B}(v_{i_t},2^{j_t}\xi r_{i_t})$. Note that since $r_{i_t}\geq T^{-C}$ this is defined.

**Lemma D.3.** *Given $t\in[T]$ with $j_t>0$ we have $\gamma_t\geq 2^{j_t-1}(\xi-1)r_{i_t}$.*

*Proof.* By definition of $j_t$ we have that $x_t\notin\mathcal{B}(v_{i_t},2^{j_t-1}\xi r_{i_t})$ so since $b_t\in\mathcal{B}(v_{i_t},r_{i_t})$ we have, by the triangle inequality, that:

$$\|x_t-b_t\|\geq\|x_t-v_{i_t}\|-\|b_t-v_{i_t}\|\geq 2^{j_t-1}\xi r_{i_t}-r_{i_t}\geq 2^{j_t-1}(\xi-1)r_{i_t}$$

Since $b_t$ is on the straight line from $x_t$ to $x_{q_t}$ we have $\|x_t-x_{q_t}\|\geq\|x_t-b_t\|$. By definition of $q_t$ we have $\|x_t-x_{q_t}\|=\Delta_{t,q_t}=\gamma_t$. Putting together gives us:

$$\gamma_t=\|x_t-x_{q_t}\|\geq\|x_t-b_t\|\geq 2^{j_t-1}(\xi-1)r_{i_t}$$

as required. $\qquad\square$

**Lemma D.4.** *For all $t\in[T]$ with $j_t=0$ we have $\gamma_t>2^{j_t-1}(\xi-1)r_{i_t}$.*

*Proof.* Since $q_t\notin\mathcal{M}$ we have $q_t\notin\mathcal{B}(v_{i_t},\xi r_{i_t})$ so since $b_t\in\mathcal{B}(v_{i_t},r_{i_t})$ we have, by the triangle inequality, that:

$$\|x_{q_t}-b_t\|\geq\|x_{q_t}-v_{i_t}\|-\|b_t-v_{i_t}\|\geq\xi r_{i_t}-r_{i_t}=(\xi-1)r_{i_t}.$$

Since $b_t$ is on the straight line from $x_t$ to $x_{q_t}$ we have $\|x_t-x_{q_t}\|\geq\|x_{q_t}-b_t\|$. By definition of $q_t$ we have $\|x_t-x_{q_t}\|=\Delta_{t,q_t}=\gamma_t$. Putting together gives us:

$$\gamma_t=\|x_t-x_{q_t}\|\geq\|x_{q_t}-b_t\|\geq(\xi-1)r_{i_t}>2^{j_t-1}(\xi-1)r_{i_t}$$

as required. $\qquad\square$

**Definition D.5.** Let $\mathcal{S}$ be a subset of $[T]$ of maximum cardinality subject to the condition that for all $s, t \in \mathcal{S}$ with $s \neq t$ we have $\Delta_{s,t} > z\gamma_t$.

**Definition D.6.** For all $i \in [N]$ and $j \in [J] \cup \{0\}$ define:

$$\mathcal{S}_{i,j} = \{t \in \mathcal{S} \mid i_t = i \wedge j_t = j\}$$

**Lemma D.7.** *For all $i \in [N]$ and $j \in [J] \cup \{0\}$ we have $|\mathcal{S}_{i,j}| \in \mathcal{O}(1)$*

*Proof.* Let $r' := 2^j \xi r_i$ and $w := z(\xi - 1)/2\xi$. By lemmas D.3 and D.4 we have, for all $t \in \mathcal{S}_{i,j}$, that:

$$\gamma_t \geq 2^{j_t - 1}(\xi - 1)r_{i_t} = 2^{j-1}(\xi - 1)r_i = wr'/z$$

so, by definition of $\mathcal{S}$, we have, for all $s, t \in \mathcal{S}_{i,j}$ with $s \neq t$, that $\|x_s - x_t\| > wr'$. Also, for all $t \in \mathcal{S}_{i,j}$ we have, by definition of $j_t$, that:

$$x_t \in \mathcal{B}(v_{i_t}, 2^{j_t}\xi r_{i_t}) = \mathcal{B}(v_i, 2^j \xi r_i) = \mathcal{B}(v_i, r')$$

So all the elements of $\mathcal{S}_{i,j}$ are contained in a ball of radius $r'$ and are all of distance at least $wr'$ apart. Since $w$ is a positive constant and the dimensionality is a constant we have the result. □

**Lemma D.8.** *We have $\Psi \in \mathcal{O}(N \ln(T))$.*

*Proof.* We have:

$$\mathcal{S} = \bigcup_{i \in [N]} \bigcup_{j \in [J] \cup \{0\}} \mathcal{S}_{i,j}$$

so that by Lemma D.7 we have $|\mathcal{S}| \in \mathcal{O}(NJ)$. Since $C$ is a constant we have $J \in \mathcal{O}(\ln(T))$ and hence $|\mathcal{S}| \in \mathcal{O}(N \ln(T))$. By definition of $\Psi$ and $\mathcal{S}$ we have that $\Psi = |\mathcal{S}|$ which completes the proof. □

**Lemma D.9.** *We have $\ln(1/\Lambda) \in \mathcal{O}(\ln(T))$*

*Proof.* Let $t$ be the element of $[T]$ that minimises $\gamma_t$ so that $\Lambda = \gamma_t$. By lemmas D.3 and D.4 we have

$$\Lambda = \gamma_t \geq 2^{j_t - 1}(\xi - 1)r_{i_t} \geq r_{i_t}(\xi - 1)/2 \in \Omega(r_{i_t})$$

so since $r_{i_t} \geq T^{-C}$ (where $C$ is a constant) we have the result. □

Since the desired bound is vacuous when $N > T$ we can assume otherwise so that by lemmas D.8 and D.9 we have:

$$\Psi \ln(1/\Lambda)^2 \in \mathcal{O}(N \ln(T)^3)$$

Substituting this into Theorem 5.2, whilst noting that $\mu_t \leq 1$ for all $t \in \mathcal{M}$, then gives us the result. ∎

# E    Proof of Theorem 4.2

Choose a set of $N' + 1$ disjoint balls of radius $2\xi r$ whose union is a subset of $\mathcal{X}$. Let the centre of the $k$-th ball be denoted $z_k$. Now partition $[T/2]$ into $N'$ sets each of size $\Theta(T/N')$. Let the $k$-th such set be denoted $\mathcal{P}_k$. For each $k \in [N']$ and $t \in \mathcal{P}_k$ let $x_t := z_k$. By applying the lower bound on the multi-armed bandit problem (see Lattimore & Szepesvári (2020)) independently to each of these sets of trials, Nature can choose some sequence of losses and a sequence $\langle \hat{y}_k \mid k \in [N'] \rangle \subseteq [K]$ such that for all $k \in [N']$ we have:

$$\sum_{t \in \mathcal{P}_k} \mathbb{E}[\ell_{t,a_t} - \ell_{t,\hat{y}_k}] \in \Omega\left(\sqrt{K|\mathcal{P}_k|}\right) = \Omega\left(\sqrt{KT/N'}\right)$$

For all $k \in [N']$ and all $t \in \mathcal{P}_k$ define $y_t := \hat{y}_k$. Summing the above equation over all $k \in [N']$ gives us:

$$\sum_{t \in [T/2]} \mathbb{E}[\ell_{t,a_t} - \ell_{t,y_t}] \in \Omega\left(\sqrt{N'KT}\right)$$

Let $\mathcal{M} := [T/2 + 1, T/2 + M]$. For all $t \in \mathcal{M}$ choose $x_t \in \mathcal{B}(z_{N'+1}, r)$ in such a way that $x_s \neq x_t$ for all $s, t \in \mathcal{M}$. It is straightforward for Nature to choose losses and a sequence $\langle y_t \mid t \in \mathcal{M} \rangle$ such that:

$$\sum_{t \in \mathcal{M}} \mathbb{E}[\ell_{t, a_t} - \ell_{t, y_t}] \geq |\mathcal{M}|/2$$

For all $t \in [T/2 + M + 1, T]$ choose $x_t := z_1$ and $y_t := \hat{y}_1$.

Note that we now have:

$$R(\boldsymbol{y}) \geq |\mathcal{M}|/2 + \Omega(\sqrt{N'KT})$$

For all $k \in [N']$ let $\mathcal{D}_k$ be the boundary of the $k$-th ball. i.e. $\mathcal{D}_k$ is the set of all $x \in \mathcal{X}$ with $\|x - z_k\| = 2\xi r$. Note that $\mathcal{D}_k$ can be covered by $\Theta(1)$ balls of radius $r$ such that none of the centres in $\{z_{k'} \mid k' \in [N']\}$ are contained in the union of these balls, when each is enlarged by a factor $\xi$.

Since $\mathcal{B}(z_{N'+1}, r) \cup \bigcup_{k \in [N']} \mathcal{D}_k$ covers a decision boundary of $\boldsymbol{y}$ we have now shown the existence of a boundary cover of cardinality $\Theta(N')$ in which each ball in the cover has radius $r$ and the only trials $t$ in which $x_t$ is in the union of the balls in the cover, when each is enlarged by a factor $\xi$, are those trials $t \in \mathcal{M}$. This completes the proof.

∎

## F  Proof of Theorem 4.3

Without loss of generality assume that $d = K = 2$. Given any algorithm we will now devise a strategy for Nature. On each trial $t \in [T]$ we shall maintain values $p_t, q_t \in [-1/2, 1/2]$ with $p_t < q_t$. We initialise with $p_1 := -1/2$ and $q_1 := 1/2$. On trial $t$ we choose:

$$x_t := ((p_t + q_t)/2, 0)$$

We then choose $y_t \in [2]$ so that $\mathbb{P}[a_t = y_t] \leq 1/2$. Next we choose $\tilde{\ell}_{t,1}$ and $\tilde{\ell}_{t,2}$ as follows. If $y_t = 1$ then $\tilde{\ell}_{t,1}$ and $\tilde{\ell}_{t,2}$ are concentrated entirely on $0$ and $1$ respectively and if $y_t = 2$ then $\tilde{\ell}_{t,1}$ and $\tilde{\ell}_{t,2}$ are concentrated entirely on $1$ and $0$ respectively. Note that:

$$\mathbb{E}[\ell_{t, a_t} - \ell_{t, y_t}] \geq 1/2$$

If $y_t = 1$ then we define $p_{t+1} := x_t$ and $q_{t+1} := q_t$. On the other hand, if $y_t = 2$ we define $p_{t+1} := p_t$ and $q_{t+1} := x_t$. Finally define:

$$\tau := ((p_{T+1} + q_{T+1})/2, 0)$$

Note that the set $\{(\tau, \sigma) \mid \sigma \in [-1/2, 1/2]\}$ is a decision boundary of $\boldsymbol{y}$. Hence the pair of balls $\mathcal{B}((\tau, 1/4), 1/4)$ and $\mathcal{B}((\tau, -1/4), 1/4)$ is a boundary cover of $\boldsymbol{y}$. Since neither of these balls contains any context in $\langle x_t \mid t \in [T] \rangle$ we have the result.

∎

## G  Proof of Theorem 4.4

The proof proceeds as in that of Theorem 4.1, but using the sharper Theorem 5.2 instead of Theorem 5.1. Since $\epsilon \leq 1/T$ the binning step moves the contexts by such a small amount that we can, without loss of generality, assume $\epsilon = 0$. Define:

$$r := (K/T)^{\frac{1}{2\varphi + 2d - \vartheta}}$$

Note that by definition of $\vartheta$ we can choose some set $\mathcal{C}$ with $|\mathcal{C}| \in \tilde{\mathcal{O}}(r^{-\vartheta})$ such that the union of the set of balls of radius $r$ with centres in $\mathcal{C}$ covers $\mathcal{D}$. Append $\mathcal{C}$ onto the end of the trial sequence (with associated losses drawn from $\nu$). Since $|\mathcal{C}| \leq T$ we can ignore the increase in the time horizon caused by this appending.

Now define the margin $\mathcal{M}$ to be the set of all $t \in [T]$ such that $x_t \notin \mathcal{C}$ but $x_t$ is within a distance of $2r$ from an element of $\mathcal{C}$. Following the proof of Theorem 4.1 we have:

$$\Psi \in \mathcal{O}(|\mathcal{C}| \ln(1/r)) \subseteq \mathcal{O}(r^{-\vartheta} \ln(T)) \tag{1}$$

Note that, since the density $\nu$ is bounded, we have that for all $x \in \mathcal{C}$ the expected number of trials $t \in [T]$ in which $x_t$ is within distance $2r$ of $x$ is in $\mathcal{O}(r^d T)$. Hence, we have that:

$$\mathbb{E}[|\mathcal{M}|] \in \mathcal{O}(|\mathcal{C}|r^d T) \subseteq \mathcal{O}(r^{d-\vartheta} T) \tag{2}$$

Now take any $t \in \mathcal{M}$. Since $x_t$ is at distance at most $2r$ from an element of $\mathcal{C}$, we have $\Delta_{s,t} \leq 2r$ for some $s \in [T] \setminus \mathcal{M}$. Hence, we have that:

$$\mu_t = \mathbb{E}[\ell_{t,y_q} - \ell_{t,y_t}] = \mathbb{E}[\ell_{t,\tilde{y}(x_q)} - \ell_{t,\tilde{y}(x_t)}] = \bar{\nu}_{\tilde{y}(x_q)}(x_t) - \bar{\nu}_{\tilde{y}(x_t)}(x_t) \tag{3}$$

for some $q \in [T]$ with:

$$\|x_q - x_t\| = \Delta_{q,t} \in \mathcal{O}(\Delta_{s,t}) \subseteq \mathcal{O}(r) \tag{4}$$

By Equation (4) we have:

$$\bar{\nu}_{\tilde{y}(x_q)}(x_t) \leq \bar{\nu}_{\tilde{y}(x_q)}(x_q) + \mathcal{O}(\|x_q - x_t\|^{\varphi}) = \bar{\nu}_{\tilde{y}(x_q)}(x_q) + \mathcal{O}(r^{\varphi}) = \min_{a \in [K]} \bar{\nu}_a(x_q) + \mathcal{O}(r^{\varphi}) \leq \bar{\nu}_{\tilde{y}(x_t)}(x_q) + \mathcal{O}(r^{\varphi})$$

which, upon substitution into Equation (3) and noting Equation (4), gives us:

$$\mu_t \leq \bar{\nu}_{\tilde{y}(x_t)}(x_q) - \bar{\nu}_{\tilde{y}(x_t)}(x_t) + \mathcal{O}(r^{\varphi}) \in \mathcal{O}(\|x_q - x_t\|^{\varphi} + r^{\varphi}) = \mathcal{O}(r^{\varphi})$$

Substituting into Equation (2) then gives us:

$$\mathbb{E}\left[\sum_{t \in \mathcal{M}} \mu_t\right] \in \mathcal{O}(r^{\varphi+d-\vartheta} T)$$

So, by combining with Equation (1) we have, by Theorem 5.2, that:

$$\mathbb{E}[R(\boldsymbol{y})] \in \tilde{\mathcal{O}}\left(r^{\varphi+d-\vartheta} T + \left(\rho + r^{-\vartheta}/\rho\right)\sqrt{KT}\right)$$

and hence, given the optimal tuning of $\rho$, we have:

$$\mathbb{E}[R(\boldsymbol{y})] \in \tilde{\mathcal{O}}\left(r^{\varphi+d-\vartheta} T + \sqrt{r^{-\vartheta} KT}\right)$$

which, noting the definition of $r$, gives us the result.

$$\blacksquare$$

