# OpenReview forum: "A Hierarchical Nearest Neighbour Approach to Contextual Bandits"
_TMLR — Accepted by TMLR_

### Review · Reviewer_SCsi · 2025-07-21

**Summary Of Contributions:**

The paper proposes an algorithm called HNN for adversarial contextual bandits, which leverages metric information across different trials to improve upon the existing HH algorithm in certain regions. The authors also demonstrate the adaptivity and low complexity of their algorithm.

**Audience:**

Yes

**Audience Explanation:**

The analysis of similarities across different trials in the study of contextual bandits offers valuable insights for real-world decision-making.

**Claims And Evidence:**

Yes

**Claims Explanation:**

The paper provides rigorous theoretical proofs and experimental studies, although some arguments raise questions. Please refer to the following issues.

**Requested Changes:**

I have some questions as follows.

1. Can $\mathcal{M}$ be linear to $T$? If so, can you provide some cases when it's sublinear? Otherwise, the bound becomes relatively weak. Similar to $\sum \mu_t$.

2. $N$ in Thm 4.1 actually contains some assumptions on the space implicitly. For general space, $N$ is not finite. So, the statement "no assumptions are made about the space itself" is not very accurate. I advise giving some examples, or classes, when $N$ is bounded.

3. On page 10, "We believe that typically we will have". Could you give some explanation or intuition on why this statement is right?

4. Since the algorithm is designed for adversarial cases, why don't you conduct some experiments to see the advantages of HNN in this situation?

5. On page 12, the statement "but are followed closely by HNN" is not accurate. The loss of HNN is almost twice the loss of NN 0.1. This result shows that the performance of HNN is not comparable to that of NN. Could you add some discussion on it?

---

> ### Author Response · Authors · 2025-08-18
>
> We thank you for your review - here are our responses.
>
> > Can $\mathcal{M}$ be linear in $T$…
>
> $\mathcal{M}$ is free to be chosen however one likes so can have any cardinality one wishes. To shed more light on how $\mathcal{M}$ can be chosen it is easiest to view what happens in euclidean space with i.i.d. stochastic contexts (i.e. how Theorem 4.4 is proved). In this case we choose $N$ balls that cover (with some overlap) the decision boundary and $\mathcal{M}$ is the set of trials whose contexts lie in the balls. Importantly, we don’t keep our choice of balls constant but as $T$ increases we choose balls of smaller radii. This means that as $T$ increases we need more balls to cover the decision boundary so $N$ increases (but not by enough to make the bound vacuous). But importantly, as the radii are smaller, $\mathcal{M}$ is sublinear in $T$. The same principle applies in the general case.
>
> > $N$ contains some assumptions…
>
> Theorem 4.1 is for the Euclidean ball. You are free to choose whichever set of balls (that cover the decision boundary) you like so $N$ can always be chosen to be finite. Our result for general spaces is Theorem 5.1 which has no assumptions at all except for bounded diameter. We note, however, that the bounded diameter is not stated in the abstract (instead being stated in the second sentence of the introduction) - we will add this.
>
> > On page 10…
>
> We do not believe that this property is always true, which is why we have used the word “typically”. In fact, the statement written here is based on an assumption that $\varphi$ and $\vartheta$ are constant throughout the space which will actually probably not be the case (although the true performance of HNN is adaptive to variation of these values - something that is not shown in the crude bound of Theorem 4.4). Because of this, we feel that we should cut this statement from the paper, which is fine as nothing depends on it. We note that this is the only point in the paper where we state a belief - everything else is proved.
>
> > Since the algorithm is designed for adversarial cases…
>
> Any experiment we can perform will be only a special case of adversarial so we chose stochastic datasets. We do note that the second experiment has a switching distribution so is not i.i.d. stochastic.
>
> > On page 12…
>
> You are right - it is not that close (by “close” we meant relative to the other algorithms) - we shall remove this phrase. We note that in the second experiment we do outperform NN. We also note that the experiments are only complementary - the main result of the paper is the attainment of the theoretical guarantees.

---

> ### Author Response · Authors · 2025-09-03
>
> We have now updated the paper to include fully non-i.i.d. experiments (to empirically show that HNN really does work well for the adversarial problem). Specifically, the raw CICIDS2017 dataset is inherently non-i.i.d. (since attacks tend to change in type over time) so we have now used the order that the data appeared in the raw dataset. HNN dramatically outperforms the other algorithms (far more than it was doing before).

---

### Review · Reviewer_usNK · 2025-08-06

**Summary Of Contributions:**

This paper studies the Contextual Bandit problem in metric spaces and proposes a new computationally efficient algorithm called Hierarchical Nearest Neighbour (HNN). The authors highlight that HNN is designed to address the "fully adversarial problem," where no assumptions are made about the metric space itself or the generation of contexts and losses. They contrast HNN with previous work, including Slivkins (2009) and Nearest Neighbour (NN) (Pasteris et al., 2023).
The major difference from the NN is that HNN introduces a hierarchical tree structure on the nearest neighbor classes and utilizes such a tree structure to reduce the computation complexity per trial.

In the bounded metric space's dimensionality setting, the author claims HNN is highly efficient, with a per-trial time complexity that is polylogarithmic in both the number of trials and the number of actions.
The author first introduces a tree updating algorithm based on the belief propagation (Delcher et al., 1995) and achieves the per-trial time complexity of $O(K\ln(T))$.
The author also mentions that HNN can be further optimized by using the CBNN algorithm (Pasteris et al., 2023) for a per-trial time complexity of
$O(\ln(K)\ln(T)^2)$.


The performance of HNN is analyzed in various settings. For the Euclidean bandit problem, the authors provide a regret bound that is based on the concept of a "boundary cover" of the comparator policy's decision boundary.
The author proves the regret upper bound of HNN in the Euclidean space is $O(N\sqrt{KT})$ (Theorem 4.1) while the regret lower bound is $\Omega(N\sqrt{KT})$.
The author compares HNN to the NN algorithm, arguing that NN can be arbitrarily bad in the mixture metric space case while HNN can still maintain a good regret guarantee.
The author analyzes HNN's performance in the i.i.d. stochastic setting, showing that it is competitive with and can sometimes outperform algorithms specifically designed for that case.
Finally, a more general regret bound is provided for arbitrary metric spaces and shows that the regret upper bound is $O(\sqrt{KT})$ (Theorems 5.1 \&5.2).

The paper includes experimental results on two real-world cyber-defence datasets: the UCI firewall dataset (int, 2019) and the CICIDS2017 intrusion dataset (Sharafaldin et al., 2018).
These two datasets are stochastic experiments.
On the UCI dataset, HNN was closely behind NN with a binning radius of 0.1, while on the CICIDS2017 dataset, HNN achieved the best performance.

**Additional Comments:**

I have several questions and hope the author can resolve them in the revision:


1. For steps 1 and 2 of HNN, is it possible that we can find multiple $s_d$ satisfying the \textit{c}-nearest neighbour criteria? If so, how to break the tie?

2. For the lower bound in the contextual bandit with a metric setting. Is it possible to include a lower bound in the general metric setting?

**Audience:**

Yes

**Audience Explanation:**

I believe some individuals in the TMLR audience would be interested in the findings of this paper, particularly due to its dual focus on theoretical guarantees and practical efficiency.

For researchers and theorists in the TMLR community, the paper's main appeal lies in its rigorous theoretical analysis of the HNN algorithm. The authors introduce a novel approach to the contextual bandit problem in metric spaces, specifically addressing the setting where no assumptions are made about the context or loss sequences. The author provides comprehensive regret bounds for HNN, first for the important special case of Euclidean spaces (Theorem 4.1) and then for general metric spaces (Theorems 5.1 and 5.2). This theoretical work is significant because it shows that HNN's performance is adaptive to the local density of contexts and the smoothness of the comparator policy's decision boundary, addressing a key shortcoming of prior work like the NN algorithm. The inclusion of an almost matching lower bound (Theorem 4.2) further strengthens the theoretical contribution.

For practitioners and engineers, the paper's findings are valuable due to HNN's computational efficiency and strong empirical performance. The algorithm is highly efficient, boasting a per-trial time complexity that is polylogarithmic in both the number of trials and the number of actions when the dimensionality of the metric space is bounded. This efficiency is crucial for real-world applications with large datasets. Furthermore, the paper provides compelling evidence of HNN's effectiveness through experiments on real-world cyber-defense datasets. The results show that HNN not only performs well but can even outperform other state-of-the-art algorithms designed specifically for the i.i.d. stochastic setting, despite being built for the more general adversarial case. This demonstrates that HNN is a robust and powerful tool that can be applied effectively in practice.

**Broader Impact Concerns:**

The Broader Impact section of the paper raises some concerns about the comparison of the HNN algorithm to existing works, both theoretically and empirically. The main point is that the submission might not have included a sufficient number of comparisons. However, the authors do compare HNN to several other algorithms, including NN, but I would like to see the author includes more popular existing works, such as SquareCB, and various classification-based methods.

A second concern revolves around the experimental evidence for HNN's performance in the adversarial setting. The paper claims that HNN is designed for this setting, but all the provided experimental results are from i.i.d. stochastic datasets. While the authors do provide theoretical proofs for HNN's effectiveness in the adversarial setting, the lack of empirical support for this specific claim could be seen as a weakness.

**Claims And Evidence:**

Yes

**Claims Explanation:**

The claims made in the submission are well-supported by both theoretical and empirical evidence. The paper uses mathematical proofs to back its theoretical claims about the algorithm's performance and efficiency. It also presents compelling experimental results from real-world datasets that demonstrate its practical effectiveness.

1. **Regret Bounds**: The author provides several regret bounds for HNN under various conditions.
    For the Euclidean bandit problem, the author presents Theorem 4.1 to prove that HNN achieves a regret bound $\|\mathcal{M}\| + O(N\sqrt{KT})$, where $|\mathcal{M}|$ is the number of history data located in the expanded boundary set. For general metric spaces, the author presents Theorem 5.1 by generalizing the covering in the Euclidean space to the packing number. Furthermore, the author gives an even more refined bound to drop the dependency on $\|\mathcal{M}\|$ in Theorem 5.2.

2. **Computational Efficiency**: HNN's efficiency has been discussed in Section 3. The author explains that HNN can achieve a per-trial time complexity of $\mathcal{O}(K \ln(T))$ in updating the posterior distribution on the action via online belief propagation.
    The above step can be improved by utilizing the CBNN algorithm to achieve a $\mathcal{O}(\ln(K)\ln(T)^2)$.

3. **Comparison to NN**: The author makes a convincing argument for HNN's superiority over the NN algorithm by using an example where the metric space is two-disjoint. The author demonstrates that NN, which uses a single binning radius, cannot adapt to variations in context density. This limitation causes a significant increase in regret, whereas HNN's adaptive nature allows it to maintain better performance.

4. **Empirical Claims and Evidence**: The paper includes empirical evidence from two experiments on real-world, i.i.d. stochastic datasets to support its claims about HNN's practical performance and show that HNN can achieve a comparable performance to NN.

**Requested Changes:**

The author claims that the HNN algorithm is designed for the adversarial problem, where no assumptions are made about the generation of contexts or losses. The experiments presented were, in fact, on datasets that are primarily i.i.d. stochastic.
While the paper does not include experiments on a purely adversarial setting, the theoretical analysis and regret bounds provided (Theorems 4.1, 5.1, and 5.2) might lack empirical evidence.
The author should include more empirical evidence to support their claim to show the performance of HNN in the adversarial contextual bandit setting.

The related work section has been expanded to include more recent and relevant contextual bandit work.
In the contextual bandit area, many existing works have both theoretical analysis and empirical evidence, such as SquareCB [1] and ILOVETOCONBANDITS [2].
It would be better to include more discussion about the position of this paper in the related work section.

To provide a clearer and more formal statement on the computational complexity, a dedicated theorem statement could be added. In this theorem statement, the author can formally state the per-trial time complexity for HNN under different implementations: When using the described belief propagation, the per-trial time complexity is $O(K\ln(T))$ under the conditions outlined in the paper.
When using the CBNN algorithm, the per-trial time complexity is $O(\ln(K)\ln(T)^2)$.

While the paper provides a detailed description of the HNN algorithm, including its core components, it would be better to give a formal algorithm block, detailing the inputs, outputs, and the step-by-step process.
An explicit algorithm block would greatly enhance clarity and reproducibility.


*[1] Beyond UCB: Optimal and Efficient Contextual Bandits with Regression Oracles*

*[2] Taming the Monster: A Fast and Simple Algorithm for Contextual Bandits*

---

> ### Author Response · Authors · 2025-08-18
>
> We thank you for your review - here are our responses.
>
> > The major difference from the NN is that HNN…
>
> Whilst HNN is indeed extremely efficient when the dimensionality is bounded, it is no more efficient than NN. The improvement over NN is the (both global and local) adaptivity to context density and decision boundary smoothness.
>
> > The author proves the regret upper bound of HNN in Euclidean space is $\mathcal{O}(N\sqrt{KT})$…
>
> This is true when the learning rate is treated as a constant, but on optimal tuning we get $\mathcal{O}(\sqrt{NKT})$ plus the margin term (i.e. the number of points under the expanded boundary cover) which almost matches the lower bound.
>
> > The author claims that the HNN algorithm is designed for the adversarial problem…
>
> Since the adversarial problem is analysed in terms of a “worst case” scenario one cannot find a dataset that can validate the adversarial performance. Any experiment we perform will only be a special case of the adversarial problem so we chose stochastic datasets. We do note that in the second experiment the dataset is not i.i.d. stochastic due to the distribution changing over time. Please can you advise us on what more you would like to see in the experiments section.
>
> > The related work section…
>
> We will expand the related work section as you have suggested.
>
> > To provide a clearer and more formal statement on the computational complexity…
>
> We will do this. We note that (when using online belief propagation) whilst the belief propagation step is $\mathcal{O}(K\ln(T))$ we still always need an additive $\mathcal{O}(\ln(T)^2)$ term to perform the nearest neighbour searches.
>
> > While the paper provides a detailed description…
>
> We will add an algorithm block.
>
> > Broader impact concerns.
>
> Please see above.
>
> > For steps 1 and 2 of HNN…
>
> Ties are broken arbitrarily.
>
> > For the lower bound…
>
> We are unsure about whether we can give a lower bound (in terms of margin and packing complexity) in general metric spaces. We will think on this further.

---

> > ### Author Response · Authors · 2025-08-23
> >
> > We have revised the paper according to the suggestions of yourself and the other reviewers. We have the following questions:
> >
> > As you requested, we have added to the positioning of our paper with the inclusion of ILOVETOCONBANDITS and SquareCB as another line of research (i.e. algorithms that reduce to supervised learning) for the i.i.d. stochastic problem. For general metric spaces, the only regression oracle (for SquareCB) that we are aware of is K nearest neighbours. We believe that KNN-UCB, which is proven to be optimal (although using different measures from, and hence having a regret bound incomparable to, our stochastic result in Euclidean space) up to log factors, theoretically outperforms SquareCB with K nearest neighbours due to the fact that K is adaptive in KNN-UCB and KNN-UCB adapts to gaps. In any case, these are i.i.d. stochastic algorithms whereas the purpose of our paper is to solve the adversarial problem. Hence, we have not discussed these works further. Is there any more that you wish us to do here?
> >
> > The purpose of this paper is to solve the fully adversarial problem which, to the best of our knowledge, has only been studied by Pasteris et. al. (2023). We have given a history of the research that led to our algorithm and have cited papers which solve the i.i.d. stochastic problem (as well as the non-i.i.d. work of Slivkins). Please do let us know if there are further steps we can take to better position our paper.

---

> ### Author Response · Authors · 2025-09-03
>
> We have now updated the paper to include fully non-i.i.d. experiments (to empirically show that HNN really does work well for the adversarial problem). Specifically, the raw CICIDS2017 dataset is inherently non-i.i.d. (since attacks tend to change in type over time) so we have now used the order that the data appeared in the raw dataset. HNN dramatically outperforms the other algorithms (far more than it was doing before).

---

### Review · Reviewer_B2Rq · 2025-08-11

**Summary Of Contributions:**

The paper studies the contextual bandit problem in general metric spaces. The main contribution is to propose and analyze an algorithm (HNN) for the adversarial setting where no assumptions are made on the space, or how the losses and contexts are generated. When the space is Euclidean, the algorithm is shown to possess regret bounds which can be better than existing methods for the adversarial setting. Regret bounds are also obtained for the i.i.d stochastic setting where the losses and contexts are drawn in an i.i.d manner from an underlying distribution. Experiments are conducted on some real datasets which show that the proposed method can outperform existing methods designed especially for the i.i.d stochastic setting.

**Audience:**

Yes

**Audience Explanation:**

The bandit problem has a long history by now, and is of considerable interest to researchers working in statistical learning theory. So this work could potentially be of interest for this community.

**Broader Impact Concerns:**

No such concerns since the work is theoretical in nature.

**Claims And Evidence:**

Yes

**Claims Explanation:**

--------------
Strengths
--------------
1. The fully adversarial contextual problem for general metric spaces has been addressed in a very limited sense in the literature, as noted by the authors. Moreover, the existing work of Pasteris et al. (2023) is not adaptive to the density of the contexts. This limitation is overcome by the proposed HNN algorithm which is adaptive to the local density of the contexts, and other quantities.

2. The experiments on two real datasets demonstrate that HNN can outperform existing methods which are designed especially for the i.i.d stochastic setting.

---------------------
Weaknesses
--------------------

1. The proposed algorithm and framework seem to rely heavily on the work of Pateris et al. (2023). In particular, the HNN algorithm is a modification of the CBNN algorithm of Pasteris et al.  which does reduce the novelty somewhat. While some effort is made to explain this, I think the main conceptual difference between the two methods could be explained cleanly somewhere to make the difference more transparent for the reader.

2. In terms of the proof techniques themselves, it is unclear how much of it relies on the mechanism developed in Pasteris et al., and what is the new component/difficulty-in-analysis, in the present paper? This could be explained more clearly in the main text, after the statement of the theorems, to make the ideas more transparent.

**Requested Changes:**

1. From the plots provided for the experiments section, it is very hard to distinguish between the different methods due to similar colour schemes, small font sizes and thin line styles. This can be improved substantially in order to help the reader check the plots in an easy manner.

2. For the stochastic i.i.d bandit problem, isn’t it a bit pessimistic to apply the HHN method – which is developed for the adversarial setting – to an easier setup?

3. In terms of the writing, and the further to the points raised earlier, I think some parts of the introduction section could be improved. For e.g., in para 2 – explain what is meant by the “i.i.d stochastic” setting. On page 2 (top), what is $p_t$? In Section 1.1, instead of having a single long paragraph which is quite hard to parse, it would be better to structure the writing into smaller paragraphs so that the reader can follow the story more easily.

4. A minor comment – the language could be toned down at certain places in the text, e.g., “…regret bound is fantastic…” on pg. 1, “…brilliant performance of our algorithm…” on pg. 7.

---

> ### Author Response · Authors · 2025-08-18
>
> We thank you for your review - here are our responses.
>
> > The proposed algorithm and framework seem to rely heavily…
>
> Rather than being a modification of CBNN, HNN uses CBNN (or online belief propagation) as a subroutine. CBNN works by receiving some $p_t\in[t-1]$ for each trial $t\in[T]\setminus\{1\}$. The novelty of HNN is the way of selecting $p_t$ (which is inspired by the bin creation subroutine of Slivkins (2009)). We note that the analysis of this method of selecting $p_t$ is substantial (see below). We will explain this better in the revised paper.
>
> > In terms of the proof techniques…
>
> We use the result of CBNN as a black box in our analysis and hence there is no overlap of our analysis and that of Pasteris et.al. (2023). We will point this out in the revised paper. Our analysis is also extremely different from (and far more involved than) that of Slivkins (2009) which was an inspiration.
>
> > From the plots provided…
>
> We will fix in the revised version.
>
> > For the stochastic bandit problem isn’t it a bit pessimistic…
>
> HNN is designed for the adversarial problem and that setting is the primary focus of the paper. However, we also wanted to demonstrate its power in the i.i.d. stochastic problem as well. Our experiments show that even in the i.i.d. case HNN can outperform algorithms designed specifically for that case.
>
> > I terms of the writing…
>
> We will incorporate your suggestions.
>
> > A minor comment - the language…
>
> We shall tone it down.

---

> > ### Comment · Reviewer_B2Rq · 2025-09-08
> > **Read the rebuttal**
> >
> > Thanks for clarifying my queries, I am satisfied with the response and have no further questions from my side.

---

### Author Response · Authors · 2025-08-23

We have revised the paper to incorporate the suggestions of the reviewers.

---

> ### Author Response · Authors · 2025-09-03
>
> We have now updated the paper to include fully non-i.i.d. experiments (to empirically show that HNN really does work well for the adversarial problem). Specifically, the raw CICIDS2017 dataset is inherently non-i.i.d. (since attacks tend to change in type over time) so we have now used the order that the data appeared in the raw dataset. HNN dramatically outperforms the other algorithms (far more than it was doing before).

---

### Decision · Action_Editor_qDBS · 2025-09-15

**Recommendation:** Accept with minor revision

**Additional Comments:**

Overall, this is a strong paper, and all three reviewers recommend accepting this paper. I agree with the reviewers to accept this paper, but I still recommend the following minor revisions:

- First, please make sure to make appropriate revisions based on the reviews.

- Second, I think the writing of some parts of this paper can be further improved, in particular: (1) in Section 3.2, please add more explanation to the belief propagation algorithm to make it easier to follow; (2) in Section 3.3, rather than saying "CBNN algorithm ... outside the scope of this paper so we refer the interested reader to...", I recommend the authors provide a high-level review of CBNN to make the paper more self-contained; (3) since this is a notation-heavy paper, I recommend that the authors add a notation table in the appendices, which will help the readers to follow this paper.

**Audience:**

Yes

**Audience Explanation:**

All three reviewers agree that some individuals in TMLR's audience would be interested in knowing the findings of this paper. After reading this paper, I agree with the reviewers.

**Claims And Evidence:**

Yes

**Claims Explanation:**

All three reviewers agree that the claims made in the submission are supported by accurate, convincing, and clear evidence. After reading the paper, I agree with the reviewers.

---

> ### Author Response · Authors · 2025-10-02
>
> Dear Action Editor,
>
> Thank you for your helpful suggestions. We have incorporated your feedback into the revised paper and, in particular, expanded the discussion of the CBNN algorithm. The revision now includes:
> - a description of the specific problem that CBNN addresses,
> - its performance guarantees (regret bound and time complexity).
>
> At the present moment, we have not given an overview of the internal mechanics of CBNN, as these are highly intricate and, we feel, cannot be meaningfully summarised without essentially reproducing the entire algorithm. Our intention is to provide readers with a clear understanding of what CBNN achieves, while keeping the paper self-contained by presenting the simpler Belief Propagation alternative in full detail.
>
> We hope this addresses your request, and we would be happy to make further adjustments if needed